# Multi-Scale Structural Design and Advanced Materials for Thermal Barrier Coatings with High Thermal Insulation: A Review

**Jinbao Song, Lishuang Wang \*, Jiantao Yao and Hui Dong**

School of Materials Science and Engineering, Xi'an Shiyou University, Xi'an 710065, China
\* Correspondence: lswang@xsyu.edu.cn; Tel.: +86-29-8838-2607

**Abstract:** Thermal barrier coatings (TBCs) are a fundamental technology used in high-temperature applications to protect superalloy substrate components. However, extreme high-temperature environments present many challenges for TBCs, such as the degradation of their thermal and mechanical properties. Hence, highly insulating, long-life TBCs must be developed to meet higher industrial efficiency. This paper reviews the main factors influencing the thermal insulation performance of TBCs, such as material, coating thickness, and structure. The heat transfer mechanism of the coating is summarized, and the degradation mechanism of the thermal insulation is analyzed from the perspective of the coating structure. Finally, the recent advances in improving the thermal insulation and lifetime of coatings are reviewed in terms of advanced materials and structural design, which will benefit advanced TBCs in future engineering applications and provide guidance for the next generation of high thermal insulating TBCs.

**Keywords:** thermal barrier coatings; high thermal insulation; coating thickness; thermal conductivity; advanced materials; multi-structural design

## 1. Introduction

Thermal barrier coatings (TBCs) are a multilayer coating of a combination of low thermal conductivity, high melting-point ceramics and alloy materials deposited on a nickel-based, high-temperature alloy to reduce the surface temperature [1–3]. TBCs are used widely to provide thermal protection on the high-temperature components of aero engines and land-based gas turbines and in the high-temperature manufacturing industry (Figure 1) [4,5]. Increasing the turbine inlet temperature can effectively improve the efficiency of gas turbines and the thrust-to-weight ratio of aero engines. As the demand for electric power and supersonic flights increases, the operating temperature of next-generation engines will exceed 1500 °C [6], which is higher than the operating temperature of nickel-based alloys. Therefore, the thermal insulation and durability of TBCs need to be improved significantly.

Typical TBCs consists of a topcoat (TC), a bond coat (BC), a thermally grown oxide (TGO), and a substrate (SUB). The functions of the TC are thermal insulation and resistance to calcium–magnesium–alumina–silicate (CMAS) corrosion. The functions of the BC are bond strengthening, oxidation resistance, and mechanical property transition [7–13]. The BC is typically a metal layer of MCrAlY (M = Fe, Ni, Co, or Ni–Co) [14–16]. The TC material selected must meet strict performance limits to resist thermal and mechanical attacks in extreme environments. These include high melting point, good strain tolerance, corrosion resistance, low sintering rate, no phase change during thermal cycling, and a coefficient of thermal expansion (CTE) that matches the metal matrix [17,18]. $Y_2O_3$-stabilized $ZrO_2$ (YSZ) is used widely as a TC material owing to its good properties [19–21]. However, YSZ undergoes a phase transformation at operating temperatures above 1300 °C, leading to volume expansion. In addition, sintering occurs, damaging the strain tolerance and

reducing the lifetime of TBCs [22,23]. Therefore, advanced TBC materials with better high-temperature stability and lower thermal conductivity are needed.

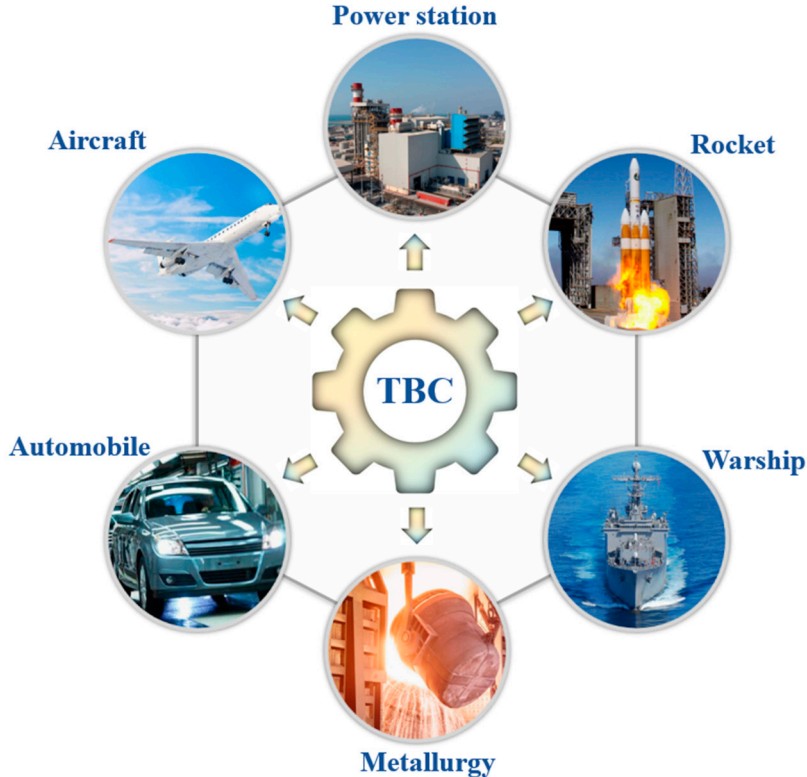

**Figure 1.** Typical applications of TBCs in industrial fields.

Conventional TBCs are mainly prepared by plasma spraying (PS) and electron beam physical vapor deposition (EB-PVD) [4]. A previous study reported that the PS-TBC coating exhibited approximately 60% lower thermal conductivity than the bulk due to the lamella structure containing many inter-splat pores [24]. The TBC prepared by EB-PVD exhibited a columnar structure with multi-scale pores. Vertical global-scale channels separate the columnar grains, while nanoscale voids are produced inside the columnar grains. The thermal conductivity decreases by less than 40% compared to that of the bulk material because the intercolumnar pores are aligned in the direction of heat flux [25]. Currently, many methods have also been developed for the preparation of TBCs; the potential and representative techniques include suspension plasma spraying (SPS) and plasma spraying physical vapor deposition (PS-PVD) (Figure 2) [26–28]. The PS-PVD process is a conventional method based on the low-pressure plasma spraying of TBCs. The increased efficiency of vacuum pumps in PS-PVD installations resulted in pressures in the working chamber between 50 and 200 Pa. Owing to the low pressure in the process, the plasma plume can reach lengths of more than 2 m and diameters of 200–400 mm [29,30]. SPS is an emerging technology that can replace atmospheric PS (APS) to manufacture thinner layers. A suspension of submicron particles can be introduced into a plasma stream via an axial or radial injection. A continuous stream of suspension flows from an injector of tens of microns and is injected into the plasma stream. The first fragmentation occurs in the plasma stream, and solvent evaporation occurs before the particles are simultaneously accelerated and melted until the particles hit the surface [31]. TBCs prepared using these two techniques have a columnar structure similar to that prepared by EB-PVD, which can improve the strain tolerance of the coatings. In addition, they have higher intracolumnar porosity, which can reduce the thermal conductivity of the coating [32], so they have been studied widely for the preparation of next-generation advanced TBCs.

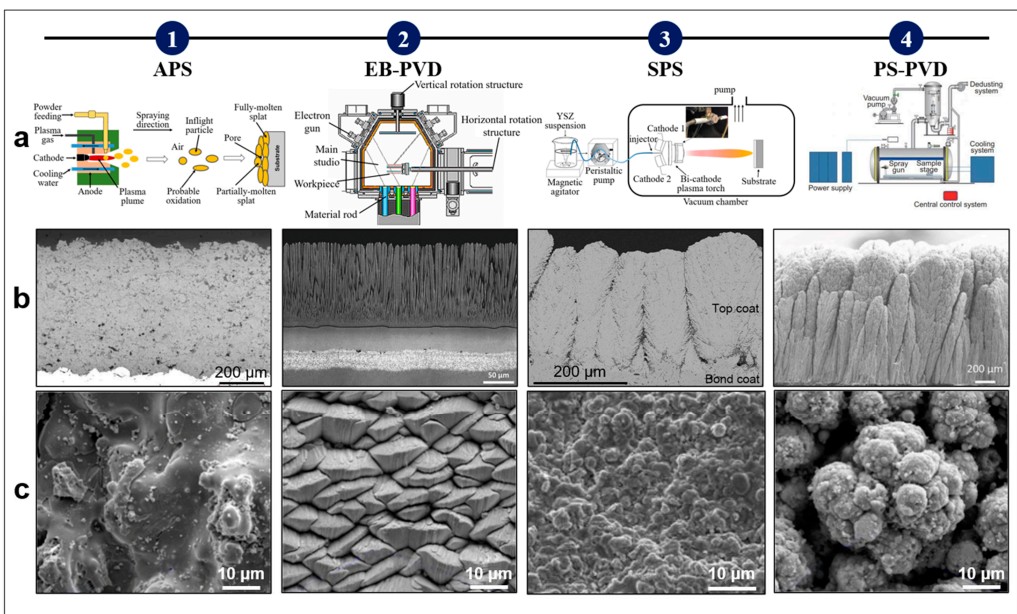

**Figure 2.** Schematic diagram of the preparation process of TBCs: (**a**) equipment working principle (**b**) cross-section (**c**) surface SEM images (adapted from [33–40]).

This paper reviews the thermal properties and influencing factors of TBCs and recent research strategies to improve their thermal insulation and durability. First, the factors affecting the thermal insulation of coatings are discussed from the aspects of material properties, coating thickness, and internal microstructure. The leading causes of thermal insulation degradation due to thermal exposure are analyzed. Finally, the related designs for high thermal insulation and long-lifespan coatings are discussed (Figure 3).

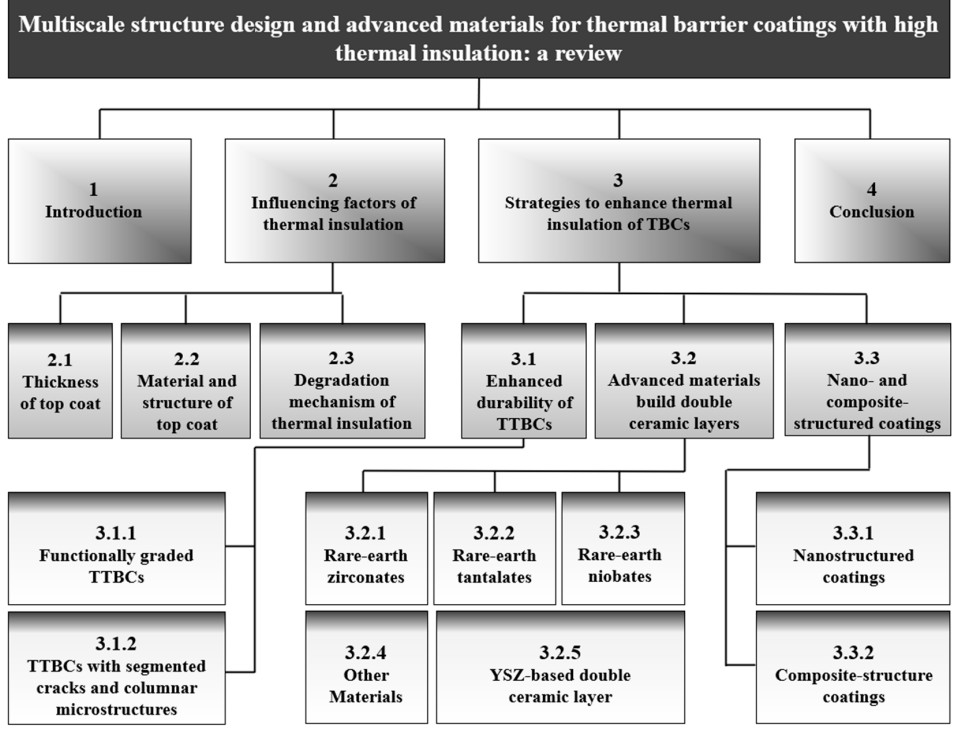

**Figure 3.** Breakdown structure for the brief review.

## 2. Influencing Factors in Thermal Insulation

The heat absorption and release of the TBC is a balanced process, and this thermal equilibrium process can be considered a steady-state heat transfer. The TBC is generally thin in the range of 200–3000 μm, and the dimensions parallel to the substrate direction can be approximated to infinity compared to the thickness direction. Therefore, the heat transfer process of a TBC can be regarded as a one-dimensional, steady-state heat transfer perpendicular to the coating surface. According to Fourier's law, the longitudinal unit thermal resistance of the coating can be expressed as follows [41]:

$$R = \frac{H}{\lambda A}, \tag{1}$$

where $H$ is the thickness of the coating, $\lambda$ is the thermal conductivity, and A is the area normal to the heat flow. Under one-dimensional, steady-state heat transfer, the thermal conductivity of the TBC depends on both the coating material and the structure. A comprehensive understanding of the heat transfer and thermal insulation degradation mechanisms of coatings is essential for the development of high thermal insulation TBCs.

### 2.1. Thickness of the Top Coat

The service environment of aviation and industrial gas turbines is different, so the requirements for TBCs on turbine blades are also different. The service time of aero-engine TBCs is thousands of hours, and it is started and stopped continuously [1], which requires a high strain tolerance TBC prepared by EB-PVD (Figure 4). Heavy-duty gas turbines need to maintain high temperatures and high efficiency for a long time, and require high thermal insulation TBCs prepared with APS [42,43]. The thickness of TBCs is often larger than that needed for aviation (Figure 5).

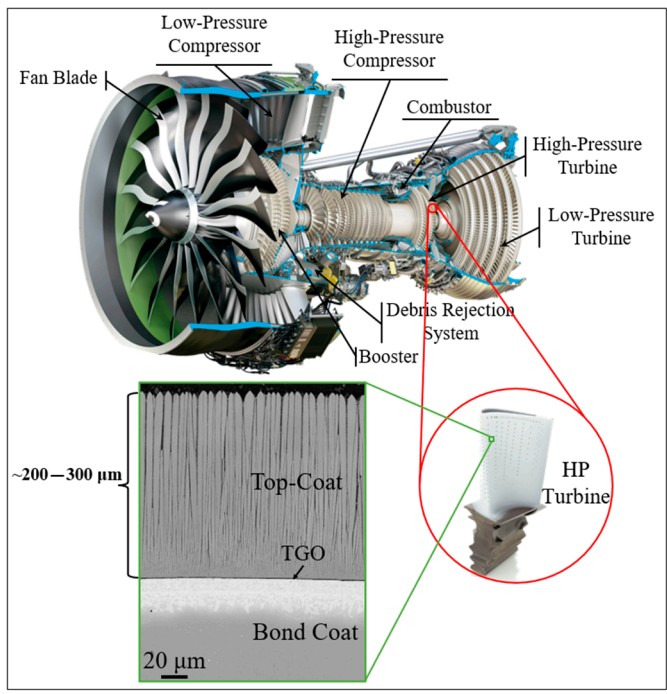

**Figure 4.** Cutaway view of aero-engine GE-9X, thermal barrier coating (TBC) on high-pressure blade, and scanning electron microscope (SEM) image of a cross-section TBC by EB-PVD 7 wt % yttria-stabilized zirconia (7YSZ). (Engine and blade images from GE aviation, and SEM micrographs adapted from Ref. [44]).

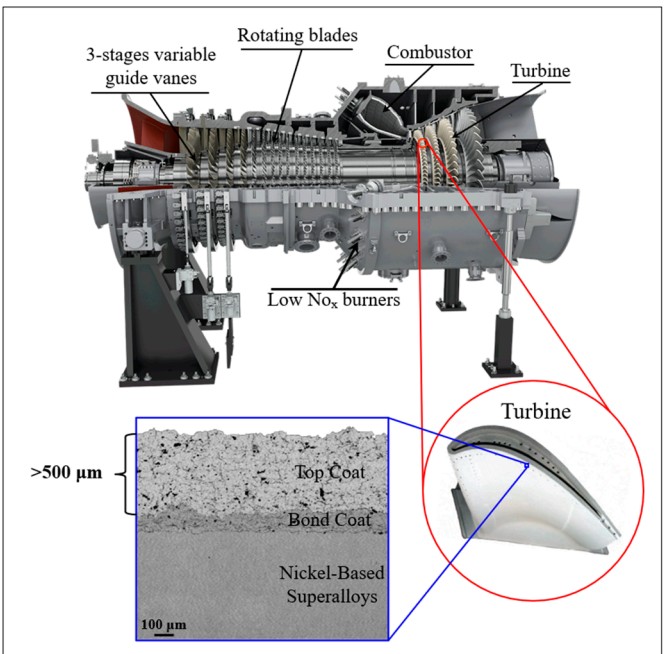

**Figure 5.** Cutaway view of gas turbine SGT5-4000F, TBC on the first stage blade, and SEM image of TBC cross-section obtained by APS 8 wt % yttria-stabilized zirconia (8YSZ). (Engine from Siemens energy, turbine images from SULZER W501F, and SEM micrographs adapted from Ref. [45]).

Increasing the thickness of the topcoat is a promising technique for improving the thermal insulation potential by preserving its mechanical properties [46,47]. However, this causes a larger thermal gradient across the coating and higher stress within the topcoat. In addition to the thermal gradient stress, a higher thickness causes the higher elastic stress energy stored within the coating, which may lead to the initiation and propagation of cracks [48–50]. For the cooling phase with a transient thermal gradient, Sundaram et al. [51] numerically determined the transient energy release rate and mode fixity for coating delamination to be dependent on the initial thermal gradient, the rate of cool down, the substrate thickness, and the substrate constraint against bending. When the substrate thickness was fixed to 3.5 mm, the energy release rate increased monotonically with the coating thicknesses of 0.15, 0.45, and 0.75 mm. Li et al. [48] investigated the effects of the thickness and mechanical load on the energy release rate of the dwelling phase. The results show that when the substrate thickness is fixed, the topcoat thickness increases, and the energy release rate increases. Mohammadzaki Goudarzi et al. [52] prepared a TBC (TC 280 μm +BC 100 μm) and a thick TBC (TTBC TC 1000 μm +BC 100 μm) using a plasma spray process and investigated their thermal insulation and durability. The scanning electron microscopy image of the polished cross-section of as-sprayed TBC and TTBC showed distributed 2D pores and globular voids (Figure 6). As shown in Figure 7, a continuous oxide scale formed at the interface of NiCrAlY/YSZ. There is no sign of horizontal cracking or debonding in the conventional TBC. On the other hand, complete delamination occurred within the TTBC, with the delamination located within the top layer near the interface. The TGO of TTBC grew at a lower rate and failed mainly due to the higher temperature drop of the thicker coating, hence the higher internal stresses at the interface between the BC and the topcoat [53]. In addition, the delamination was driven by the high elastic strain within the topcoat caused by the higher thickness [46]. The thermal insulation of TTBC was two times that of a TBC. However, the life and bond strength of the TTBC were only approximately 43% of a TBC (Figure 8). Liang et al. [54] examined the effects of the coating thickness on fracture under thermal shock cycling by combining their thermal shock experiments with the corresponding finite element analysis. The results suggested that coatings with thicknesses greater than 300 μm were prone to failure after

fewer cycles, and the coating interface cracking was more significant. The 500 μm coatings failed at only 16.6% of the number of cycles for the 100 μm coatings; the potential failure mechanism is the higher compressive stress of thicker coatings. Vo et al. [55] investigated the effect of the TBC thickness on realistic high-pressure turbine cooling-blade heat transfer using high-quality meshes based on the Mosaic meshing technique. The results showed that the blade surface temperature decreases with increasing TBC thickness (Figure 9). The blade coated with a 0.2 mm thick TBC reduced the blade surface temperature by 230 K (from 1150 to 920 K) compared to the blade without the TBC coating. The TBC thickness increased by 0.2 mm, resulting in an approximately 40 K decrease in the maximum blade temperature. Under the existing conditions, 250 μm thick TBCs can reduce the surface temperature of the blade substrate by 384–440 K [56]. The TBC thickness is significant for preventing heat transfer to the blade surface. However, spallation of TTBCs during gas turbine engine operating conditions is one of the major problems in the development and application of these coatings.

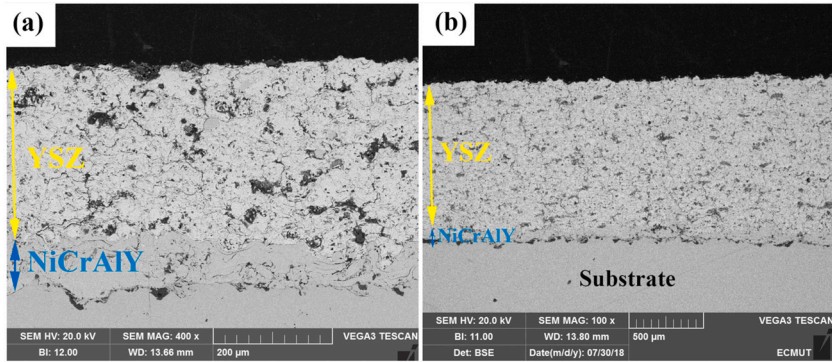

**Figure 6.** SEM images of the cross-sectional area of different thicknesses for a topcoat of (**a**) 280 μm (**b**) 1000 μm [52].

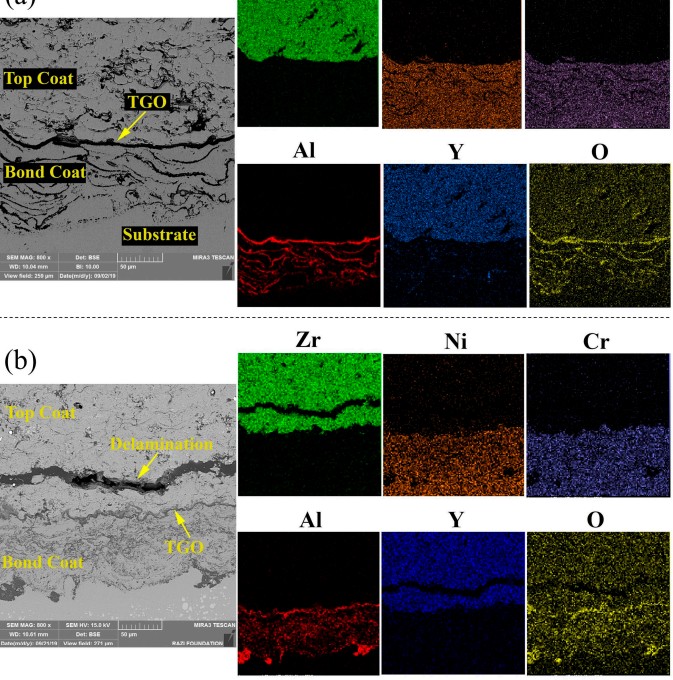

**Figure 7.** SEM image and EDS analysis map of the polished cross-section of the TBC with the topcoat of (**a**) 280 μm (**b**) 1000 μm after 150 h of oxidation test at 1000 °C [52].

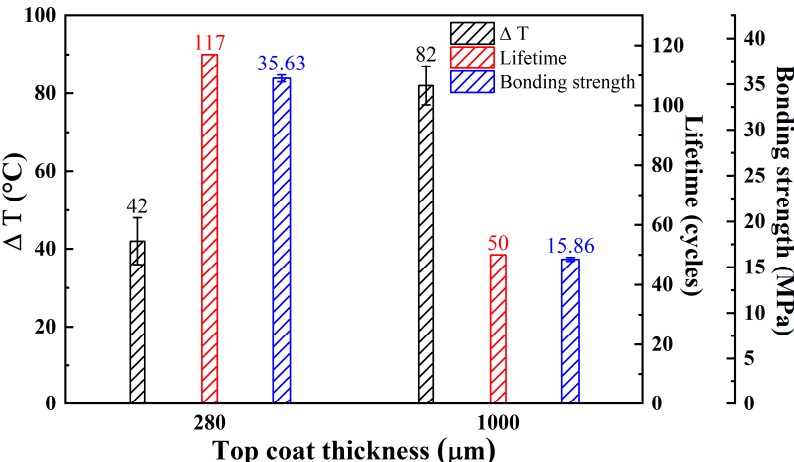

**Figure 8.** Effect of topcoat thickness on coating thermal insulation (ΔT is the difference between the surface temperature of the uncoated substrate and the surface temperature of the substrate coated with the TBC), thermal shock lives (at 1000 °C), and bonding strength. (adapted from [52]).

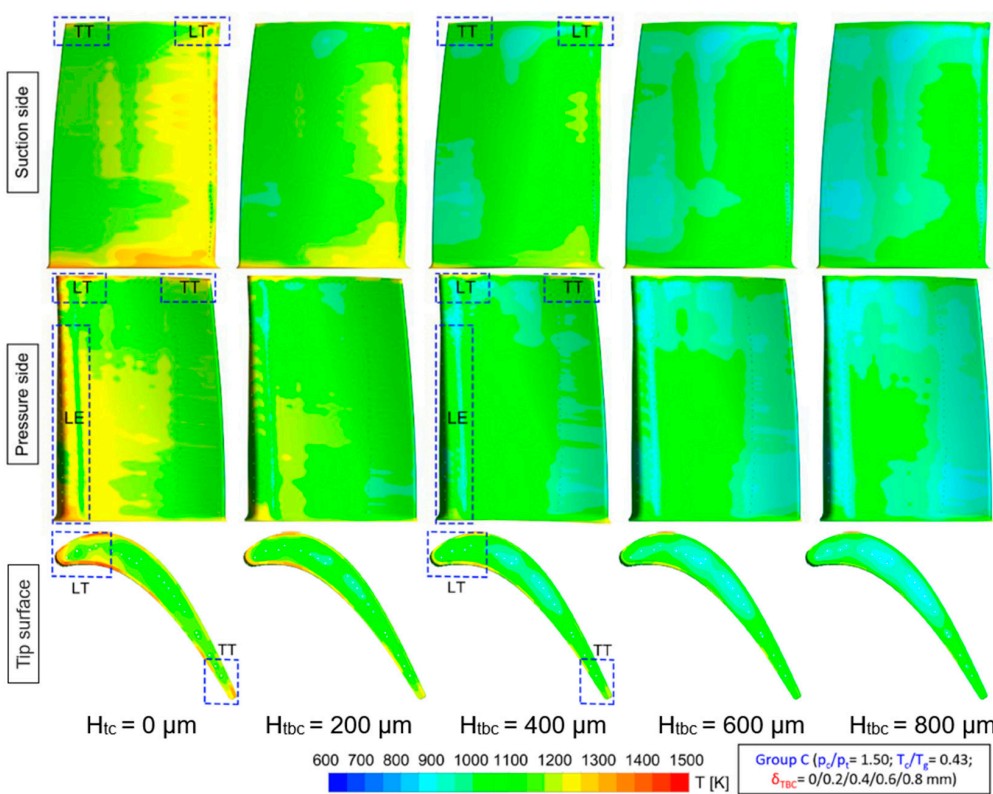

**Figure 9.** Comparison of temperature on the blade surface and the tip surface [55]. ($H_{tc}$ is thickness of the topcoat, LE, LT, and TT are leading edge, leading tip, and trailing tip, respectively).

## 2.2. Materials and Structure of Top Coats

The heat transfer mechanism in conventional TBCs consists mainly of phonon heat conduction (crystal lattice vibration) within the YSZ material, photon heat transfer (thermal radiation), and gas-phase heat conduction and Knudsen heat transfer within the gas pores [57]. For electrically insulating ceramic materials, electrons do not contribute to the heat transfer process. For TBCs in the medium- to high-temperature range, the heat transfer process of the

material is mainly phonon transport [58,59]. Based on the Debye theory, the contribution of lattice vibrations to heat conduction according to phonon dynamics theory is:

$$K_p = \frac{1}{3} \int C_v \rho v l_p, \tag{2}$$

where $C_v$ is the specific heat capacity per unit volume of the material, $\rho$ is the density, $v$ is the phonon velocity, and $l_p$ is the phonon mean-free range. $C_v$, $\rho$, and $v$ are influenced mainly by the type of material. The phonon mean-free range $l_p$ is the parameter with the greatest influence on the thermal conductivity of TBCs. The contribution of phonons to heat transfer is inversely proportional to the mean-free range of phonon scattering, which is closely related to the microstructure of the material. The scattering processes of phonons can be generally classified into three main categories: phonon-boundary scattering, phonon-defect scattering, and phonon–phonon scattering. According to the Matthiessen rule, the effects of the three scattering processes on the mean-free range of phonons can be quantified as follows:

$$\frac{1}{l} = \frac{1}{l_{\text{defect}}} + \frac{1}{l_{\text{boundary}}} + \frac{1}{l_{\text{phonon}}}, \tag{3}$$

where $l_{\text{defect}}$, $l_{\text{boundary}}$, and $l_{\text{phonon}}$ denote the effect of defects, boundary, and other phonons on the phonon mean-free range, respectively. A large number of defects and pore boundaries inside the TBC (Figure 10) affect the average free range of its internal molecules. The enhanced scattering of particles significantly affects the thermal conductivity of the coating [60]. After thermal exposure of TBC, the average free range of phonons increases. Hence, the proportion of photon heat transfer increases, and the contribution of photon heat transfer to thermal conductivity can be expressed as:

$$K_r = \frac{16}{3} \sigma n^2 T^3 l_r, \tag{4}$$

where $\sigma$ is the Stephen–Boltzmann constant, $n$ is the refractive index, $T$ is the absolute temperature, and $l_r$ is the photon scattering mean-free path. When the temperature is lower than 1200 °C, YSZ transfers heat mainly by phonons, but when the temperature increases, the proportion of photon heat transfer becomes larger (10% of the total heat transfer at 1250 °C). Therefore, for future high-performance engines, thermal radiation is an important factor affecting their performance and efficiency. The lamella structure of the PS-TBC can reduce the radiation heat transfer of TBCs.

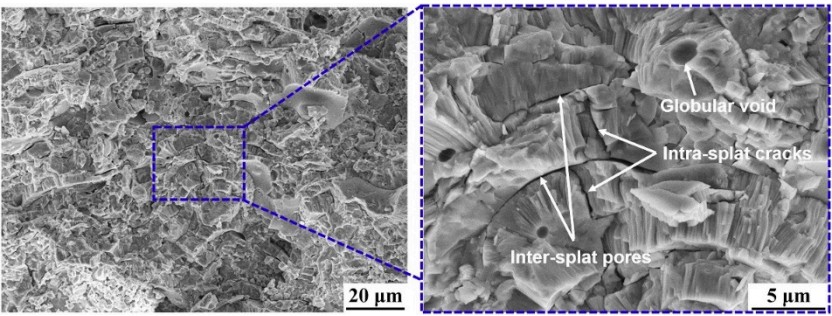

**Figure 10.** Fractured cross-section of as-deposited PS-TBC [61].

The thermal conductivity of gas inside the pore in TBC can be calculated using the $K_g$ model for the thermal conductivity of a gas inside a restricted air channel with a characteristic length $d_v$, which can be calculated using the following equation [62]:

$$K_g = \frac{K_g^0}{1 + BT/(d_v P)}, \tag{5}$$

where $P$ is the pressure, $T$ is the absolute temperature, $K_g^0$ is the unconstrained conductivity of a gas at the temperature concerned, and $B$ is a constant that generally depends on the gas type and the solid surface material, surface roughness, and gas–solid interactions [63]. Indeed, the magnitude of the gas pressure $P$ and the gas temperature $T$ directly affect the molecular mean-free range $\lambda$ of the gas inside the pore, significantly affecting the thermal conductivity of the gas phase inside the pore and its thermal conductivity. At atmospheric pressure, the molecular mean-free range $\lambda$ is larger than the characteristic length $d_v$ of the pore, and Knudsen heat transfer occurs in the pore, which causes the thermal conductivity $K_g$ of the gas inside the pore to be much lower than that of the free gas at the same temperature. $K_g$ barely changes with temperature and does not increase with temperature similar to the free-space gas thermal conductivity $K_g^0$, which is beneficial to the overall thermal insulation performance of the TBC. However, under the high-pressure conditions of a TBC, the average free range of molecules will be smaller than the characteristic length of the pore, and the thermal conductivity of the gas in the pore will be larger than that of the gas in the pore at atmospheric pressure. Moreover, the thermal conductivity of the gas in the pore will increase with increasing temperature. The value will be close to the thermal conductivity of the gas in the free space with increasing pressure, which will adversely affect the overall thermal insulation property of the TBC.

Many traditional empirical formulas have been proposed to calculate the effective thermal conductivity of coatings [64–68]. Neuer et al. [69] studied the effect of the porosity of YSZ coatings on the thermal insulation performance and reported that the relationship between the porosity and thermal conductivity of TBCs is:

$$\lambda_p = \lambda_0 (1 - \beta \cdot p), \tag{6}$$

where $\lambda_0$ is the thermal conductivity of the bulk material, $\beta$ is a constant, and $p$ is the porosity of TBCs. A simulation of the geometric arrangement and heat flows through it is required to incorporate the effects of heat transfer through the gas in the pores and within the zirconia itself and the pore architecture [70,71]. Golosnoy et al. [72] established analytical and numerical models for simulating heat flow through a two-component structure consisting of a solid layer separated by thin, periodically bridged, gas-filled voids, designed to represent plasma-sprayed ceramic coatings. It is based on dividing the modeling domain into two regions: (1) unidirectional conduction successively through the splits and the intervening air gaps and (2) channel conduction through bridges (bonding area). The results predict that the bridge area is the microstructural parameter with the greatest impact on the conductivity. Several factors play significant roles in determining the thermal conductivity, such as pore size, pore distribution, and crack orientation [73–81]. Wei et al. [82] developed an analytical model to discuss the effects of the microstructural parameters, including splat thickness, bonding ratio between splits, and unit size, on the total thermal resistance and to reveal the dominant effect of oriented 2D pores on heat flux (Figure 11) [72,82,83]. The thermal conductivity is strongly associated with the 2D composited stacking structure: bulk splits and air-trapped pores, and microstructural parameters, including splat thickness, splat/splat bonding ratio, and splat length contribute to total thermal resistance.

Higher porosity corresponds to lower thermal conductivity. However, at the service temperature, the radiation heat transfer caused by the pores cannot be ignored [84–88]. Numerous studies have shown that pores parallel to the direction of the substrate can drastically reduce the thermal conductivity [41,62,67,73,89–91]. Boissonnet et al. [92] prepared a series of TBCs with different thicknesses by PS and high-velocity oxygen-fuel spraying (Figure 12). They reported that the thermal diffusivity values decreased linearly with increasing lamellar porosity. The dependence of the thermal diffusivity on the porosity decreased at high temperatures (Figure 13). Sun et al. [73] investigated the thermal conductivity of various TBC pore morphologies by numerical methods. The thermal conductivity decreased as the total porosity increased, and the TBC with pores in the 0° direction (vertical to heat flow) had the lowest thermal conductivity. Furthermore, from 0° to 60° in the pore direction, the thermal conductivity decreased as the pore aspect ratio increased

(Figure 14). In summary, the porosity level affects the thermal conductivity of the coating, while the pore morphology, aspect ratio, and temperature further influence the dependence of porosity and thermal insulation of the coating.

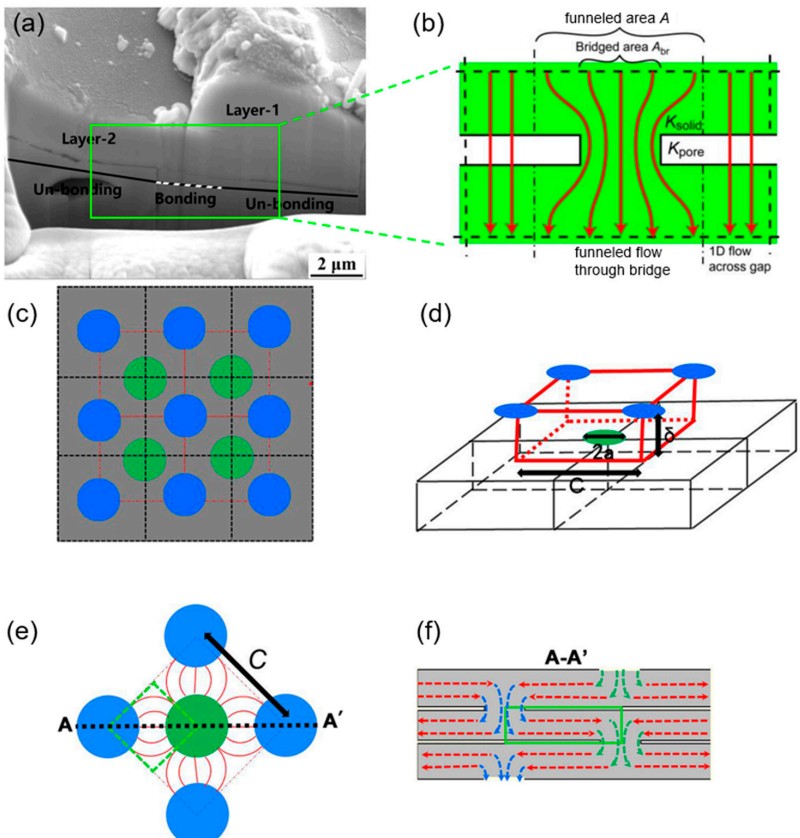

**Figure 11.** (**a**) Partially bonded layers of a plasma-sprayed coating prepared by focused ion beam (**b**) schematic representation of the "two flux regimes" model for heat flow through a set of plates connected via bridges. Analytical model to investigate the dominant effect of 2D pores on the prevention of heat flux: stacking of unit splat segments (**c**) top view (**d**) perspective view (**e**) top view of a basic stacking pattern (**f**) cross-section of A-A′ corresponding to (**e**).

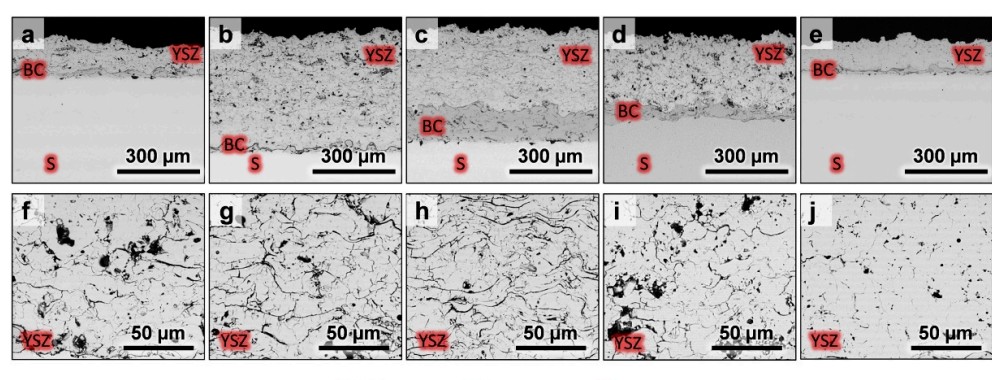

NB: **YSZ** = ceramic; **BC** = bond coating; **S** = substrate

**Figure 12.** SEM micrographs of cross-sections of TBCs with different parameters: (**a**) AM1-APS 1: Haynes 188 + 40 μm NiCrAlY + 110 μm YSZ, (**b**) AM1-APS 2: Haynes 188 + 15 μm NiCrAlY + 430 μm YSZ, (**c**) DFL-APS:Haynes 188 + 130 μm NiCoCrAlY + 330 μm YSZ, (**d**) In-APS: Inconel 600 + 72 μm NiCrAlY + 230 μm YSZ, (**e**) AM1-APSmf: Haynes 188 + 15 μm NiCrAlY + 90 μm YSZ, and (**f**–**j**) are the corresponding higher magnifications of the ceramic coatings [92].

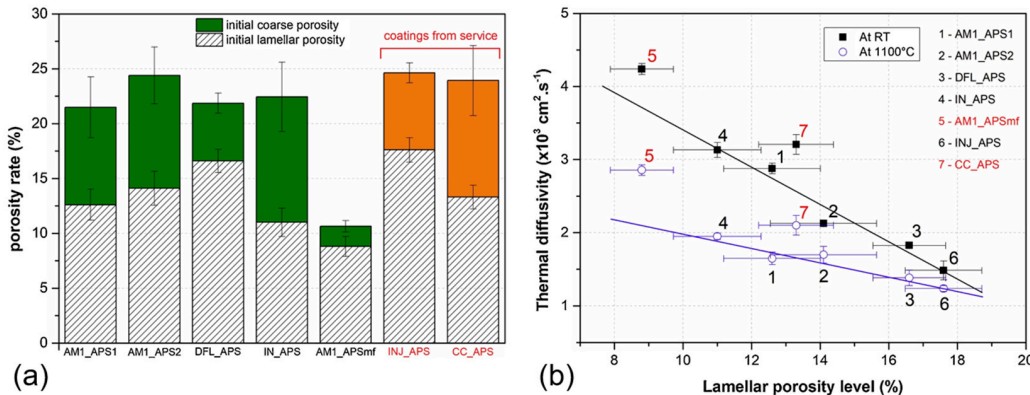

**Figure 13.** (**a**) Porosity and (**b**) thermal diffusivity of TBCs, (INJ-APS: Haynes 188 + 70 μm NiCoCrAlY + 450 μm YSZ, CC-APS: Haynes 188 + 135 μm NiCoCrAlY + 190 μm YSZ, other group parameters are consistent with Figure 9 [92].

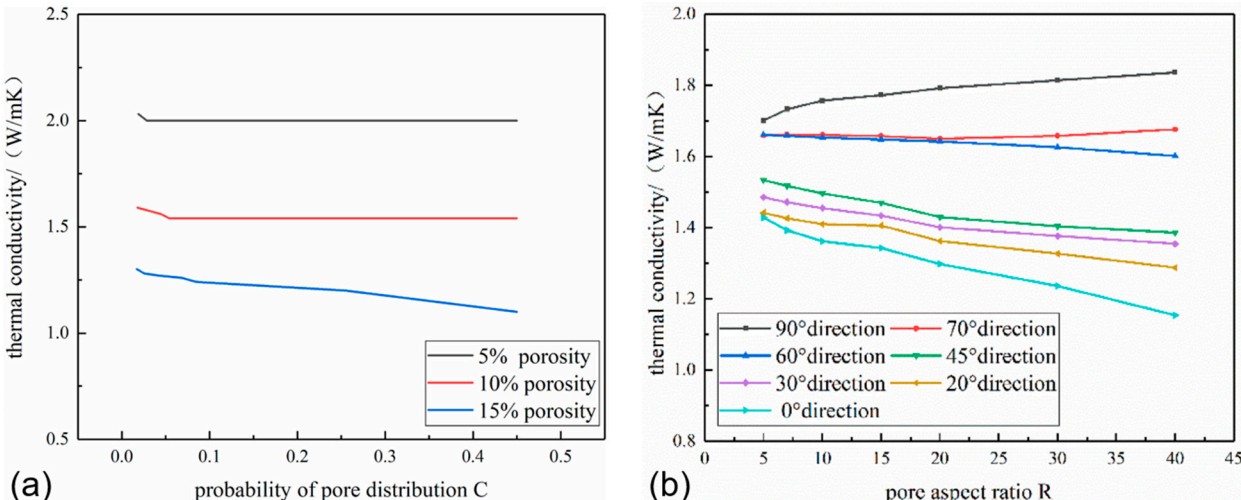

**Figure 14.** The evolution of thermal conductivity (**a**) with pore sizes at different total porosities (**b**) with pore angles at different pore aspect ratios [73].

Therefore, the inherent thermal conductivity of the bulk material and the thickness and multi-scale pore structure of the coating are the key factors in improving the thermal insulation performance of TBCs [41,93–95].

### 2.3. Degradation Mechanism of Thermal Insulation

The thermal conductivity of TBCs increases significantly after thermal exposure, resulting in a decrease in the thermal insulation of the coating. Boissonnet et al. [92] suggested that the thermal diffusivity of self-standing TBCs and full TBCs increases by approximately 40% after thermal exposure (Figure 15), but the self-standing thermal diffusivity first increased, then decreased, and then stabilized because of the generation of many cracks during cooling. Cernuschi et al. [96] reported a similar increase in thermal diffusivity with heat treatment time. A higher temperature resulted in a more severe increase (Figure 16). The thermal conductivity of coatings becomes much more complicated in actual service. The main problem is that the evolution of the microstructure of TBCs by thermal service causes a change in the thermal conductivity of the coating [97–99].

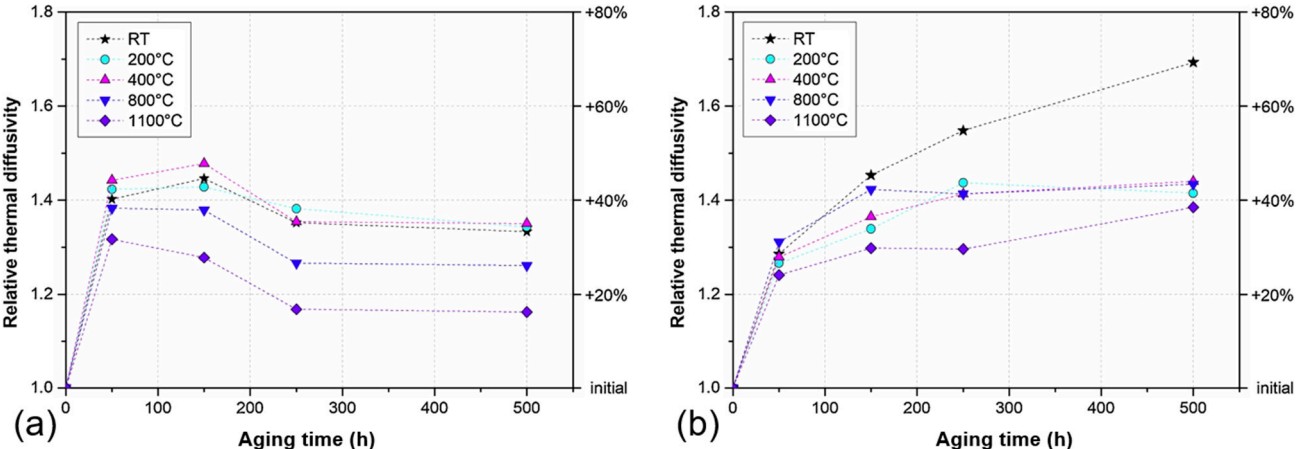

**Figure 15.** Normalized thermal diffusivity of TBCs as a function of isothermal heat treatment time at 1100 °C (**a**) freestanding coatings (**b**) full TBCs [92].

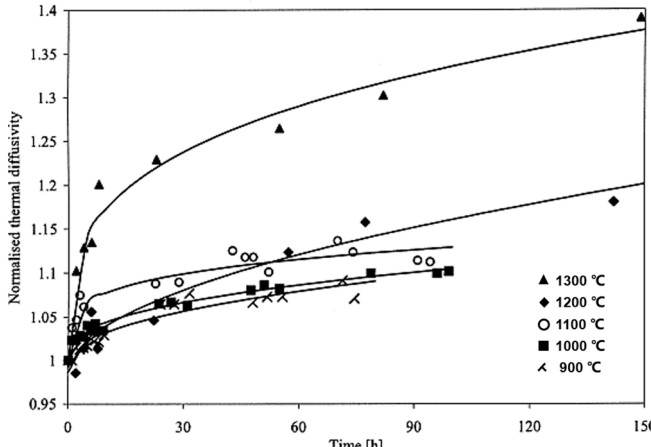

**Figure 16.** Normalized thermal diffusivity of TBCs at different temperatures as a function of thermal exposure time [96].

Sintering leads to an approximate 80% increase in thermal conductivity [77,98,100,101]. Many studies attributed this to the significant healing of oriented 2D pores [85,101–107]. Li et al. [23] reported the comprehensive sintering mechanism for lamellar TBCs. An overall property evolution with two-stage kinetics was presented during thermal exposure. The first stage (0–10 h) resulted in faster sintering kinetics and increased mechanical properties due to the rapid healing caused by multipoint connection at the inter-splat pore tips, as well as a small quantity of narrow intra-splat cracks.

A previous study [61] reported the dynamic evolution of a coating sintered at high temperatures and the healing mechanism of 2D pores. Figure 17 shows the structural changes to the two types of pores with different widths during thermal exposure. The grains begin to grow within a relatively short time after thermal exposure, and the upper and lower surfaces of the pores become rough. Several diffusions short-circuit channels are formed in the pores, accelerating the high-temperature evolution typical of pore healing (Figure 17a). In addition, the wider 2D pores are not completely healed after the same thermal exposure (Figure 17b), and still maintain the narrow and long 2D features. However, the point of contact in the narrow area separates the 2D pores into several small pores, which resulted in a reduction in the aspect ratio of the 2D pores, thereby impairing the thermal insulation properties of the coating. Liu et al. [103] characterized the evolution of interlamellar 2D pores of the APS LZO coatings during exposure to high temperatures for different durations. The aspect ratio of the interlamellar 2D pores in the as-sprayed coating

was more than 50. As the exposure duration increased, grain bridging segmented long gaps into several regions and reduced most pores to smaller ones with smaller aspect ratios. This led to a rapid decrease in the aspect ratio of the residual 2D pores at the early stage. The aspect ratio of most residual 2D pores after five hours of thermal exposure became less than 30. After thermal exposure, the aspect ratio was distributed mainly from 10 to 30. This sintering effect increases the stiffness and thermal conductivity of TBCs (Figure 18). Therefore, the healing of 2D pores and the increase in the aspect ratio at high temperatures are the main reasons for the increase in the thermal conductivity of TBCs [77,98,103,108].

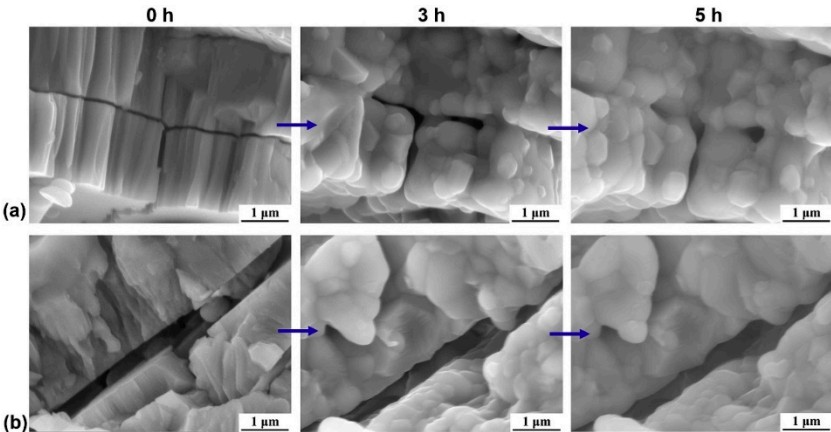

**Figure 17.** In situ healing of 2D pores of APS-TBC during thermal exposure: (**a**) a pore of less than 50 nm width (**b**) a pore of approximately 200 nm width [61].

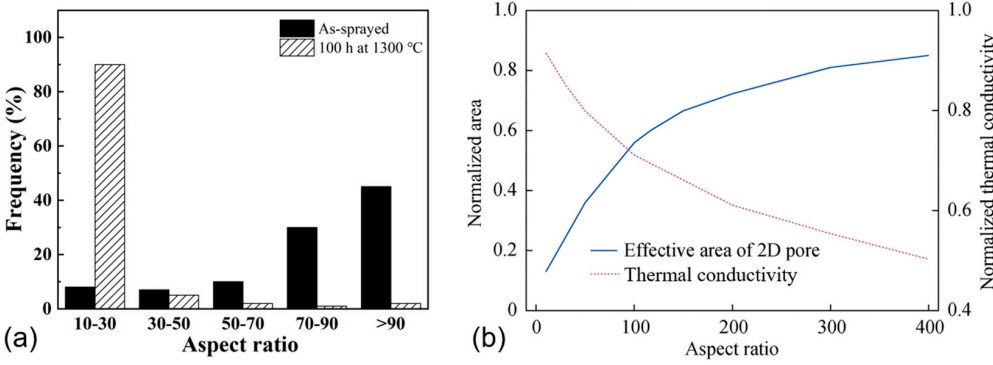

**Figure 18.** (**a**) Aspect ratio distribution of interlamellar 2D pores in APS-TBC exposed at 1300 °C for different durations (**b**) relationship between the pore normalized area, normalized thermal conductivity, and aspect ratio (adapted from [103,109]).

### 3. Strategies to Enhance the Thermal Insulation of TBCs

The thermal insulation and lifetime of TBCs curb the efficiency and reliability of the gas turbine. According to the previous section, the improved thermal insulation of TBCs can be achieved by increasing the thickness of the ceramic layer, reducing the thermal conductivity of the ceramic material, and adjusting the multi-scale structure of the coating. However, ceramic layer thickness and low thermal conductivity of ceramic materials subjected to thermal stress and fracture toughness affect the lifetime of TBCs. Combining with a multi-scale structural design approach may be an effective way to solve this problem. Some potential ceramic materials and structural designs are summarized, which will guide the development of high thermal insulation and long life TBCs.

### 3.1. Enhanced Durability of TTBCs

TTBCs (>500 μm) are effective in thermal insulation [47,49,54,110,111]. Owing to the difference in thermal expansion coefficients between TCs and the substrate, a greater TC thickness indicates a greater thermal gradient within the coating and thermal mismatch stresses, elastically stored strain energy, and the energy release rate for crack formation, resulting in lower bonding strength and faster TTBC degradation [53,112–117]. Furthermore, the singularity of the stress at the coating-free edge also influences the failure of the coatings [118]. The broken regions usually occur at the horizontal and vertical cracks within the TC and TC/BC interfaces [17,119–123].

As the thermomechanical properties of TBCs are strongly dependent on the microstructure of the coating [61,75,93,124,125], the structure can be optimized in terms of the chemical composition, pore and crack distribution to achieve high thermal insulation and a long lifespan. Some methods to tailor the microstructure have been reported, including segmentation-cracked coatings, porous coatings, dense coatings, columnar coatings, gradient coatings, or multilayer coatings, by optimizing the spraying parameters or adopting various powders [50,126–132]. TTBCs exhibit a range of properties according to their structures.

#### 3.1.1. Functionally Graded TTBCs

The tendency of the internal cracks of a TC to extend due to poor fracture toughness is a major problem [128]. Many researchers have fabricated gradient-structured YSZ-TBCs, which appeared to be a good option for improving the fracture toughness near the TC/BC interface of a TTBC [133–135]. Gradient coatings can reduce interfacial stress and increase fracture toughness [127,128]. The gradient structure has two forms: structural gradient and component gradient. The spraying power and gas flow rate altered the morphology of the splits, resulting in different coating structures obtained by subsequent deposition [136]. Functionally graded TBCs were sprayed by varying the feeding ratio of YSZ/NiCrAlY powders.

The porosity and pore shape are the key parameters controlling the mechanical properties and sintering behavior of TBCs [137,138]. Low porosity is beneficial to the fracture toughness of the coating, and high porosity is beneficial to maintaining strain tolerance. The functional structure classification can provide a coating with different microstructures on the longitudinal scale to give the coating strong integrated mechanical properties. The graded coating releases the driving force energy to fail while preventing the crack length [139]. To improve the fracture toughness of the TC layer near the BC, Li et al. [129] designed three layers of TBCs with different microstructures. The layer adjacent to the BC has continuous columnar grains to achieve crack extension resistance. The upper layer uses conventional and porous coatings to transition and improve the thermal insulation, respectively. This can increase the thermal cycle life of the coating by approximately 3.5–4 times that of conventional coatings. Lv et al. [140] prepared TBCs with gradient pores and examined their mechanical and thermal properties (Figure 19). The coatings with decreasing porosity from top to bottom showed improved sintering resistance, and the coating demonstrated a compressive stress state at the interface (Figure 20), indicating favorable delamination resistance and long lifetime. Structurally graded coatings allow the microstructure to be adjusted to the functional needs of the layer, resulting in an excellent combination of mechanical properties.

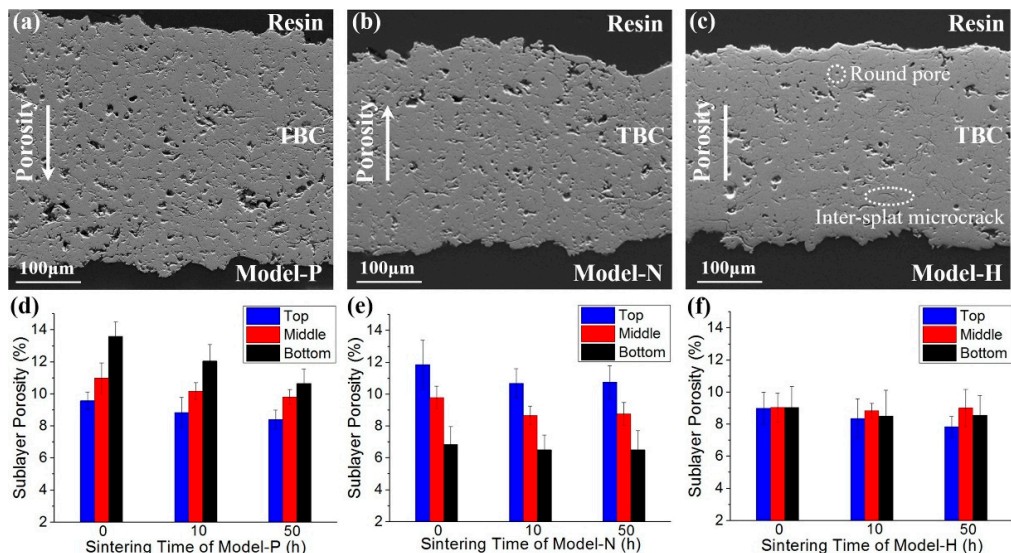

**Figure 19.** SEM cross-sections of as-sprayed, free-standing TBCs with different porosity gradients: (**a**) positive gradient (Model-P) with increasing porosity from top to bottom (**b**) negative gradient (Model-N) with decreasing porosity (**c**) homogeneous porosity (Model-H) with uniform porosity distribution; porosity distributions of top, middle and bottom sublayers at different sintering time in (**d**) Model-P (**e**) Model-N (**f**) Model-H [140].

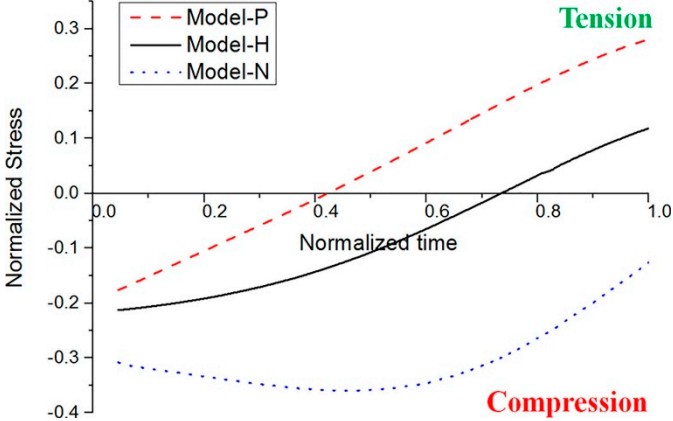

**Figure 20.** Evolution of stress state at the bottom of graded and homogenous porous coatings in the graded temperature field [140].

The composition-graded coating from 100% metal on the substrate to 100% ceramics on the top layer caused the interface between the two substances to disappear [141] while retaining the properties of a variety of composite materials. The composition-graded coatings improve the bond strength and toughness of the coating and reduce the thermal stress at the interface [142–145]. Saeedi et al. [146] prepared a five-layer functionally graded APS-TBC by changing the composition of YSZ/NiCrAlY. Cracks formed in much lower quantities compared to the conventional TBC owing to the accommodation effect of the functionally graded structure on thermal stress. In addition, the bond strength of the gradient functional coating was approximately 1.5 times that of conventional TBCs. Mohammadzaki Goudarzi et al. [147] examined the effects of composition and porosity gradients on the thermomechanical properties of APS-TTBCs. They prepared a functionally graded TTBC (FGTTBC) by changing the hydrogen flow rate to prepare the top layer with gradient pores and mixing different percentages of NiCrAlY and YSZ powders (Figure 21). Four samples with a total TC thickness of approximately 1100 ± 40 μm were prepared for the performance comparison, namely, conventional TTBC, FGTTBC, FGTTBC porous

(FGTTBC-P), and FGTTBC-pore gradient (FGTTBC-PG) (Figure 22). The results also showed that a functionally graded coating has a good bond strength. Compared to TTBCs, the fracture toughness of the FGTTBC was increased by 28%, but its thermal insulation was the worst because of the presence of a higher thermal conductivity material (NiCrAlY) in TCs. FGTTBC-P has the best life and thermal insulation because of the high porosity and low elastic modulus (Figure 23). The presence of NiCrAlY in TCs improves the total coating fracture toughness and crack growth resistance owing to crack bridging in a graded-volume fraction [135,148,149]. A porous structure improves the thermal insulation of the coating.

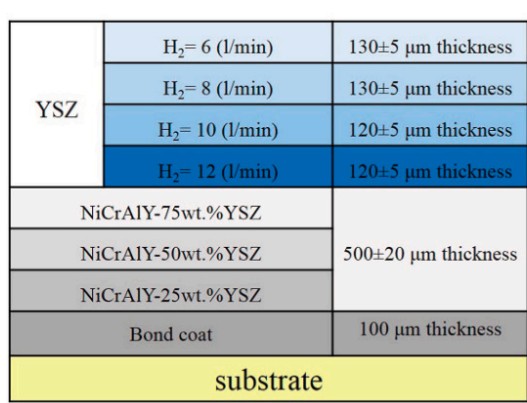 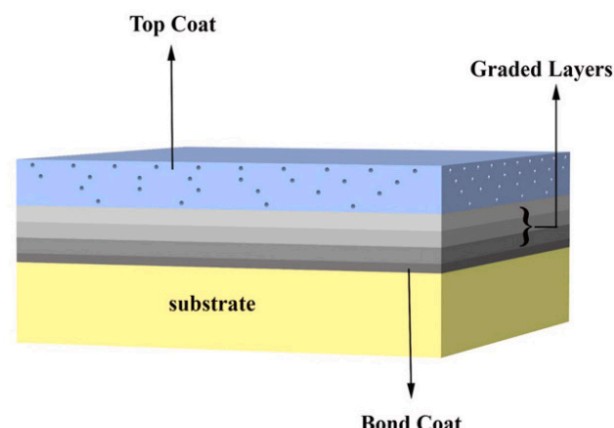

**Figure 21.** Schematic illustration of how the layers are arranged in the FGTTBC-PG and the parameters used to create these layers [147].

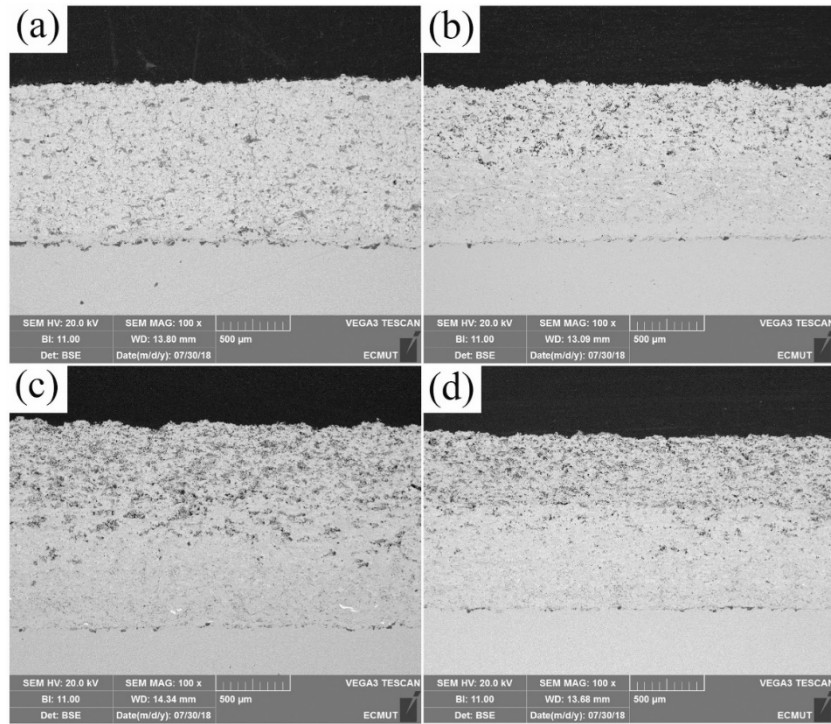

**Figure 22.** SEM images of the sample cross-section: (**a**) TTBC (**b**) FGTTBC (**c**) FGTTBC-P (**d**) FGTTBC-PG [147].

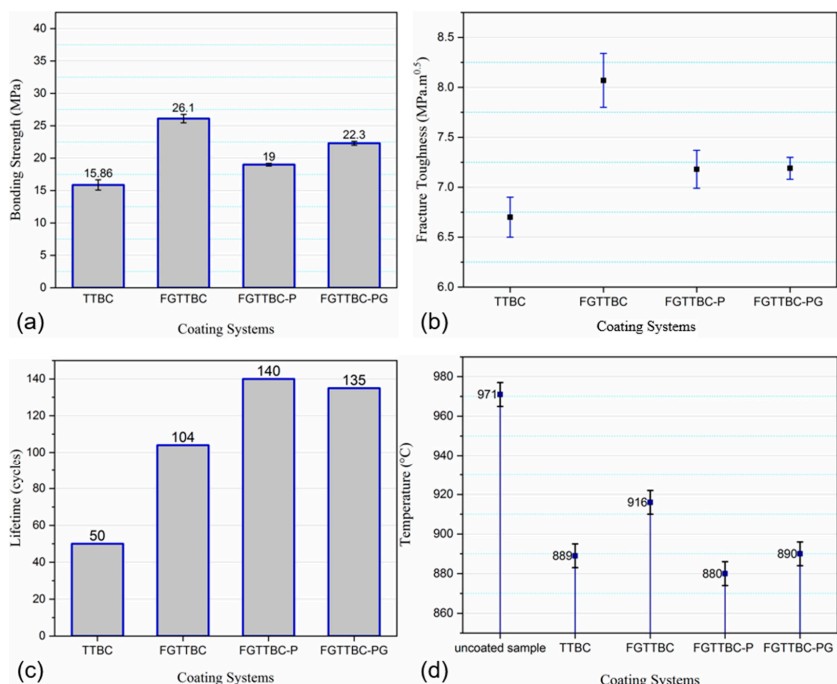

**Figure 23.** (**a**) Bonding strength (**b**) the fracture toughness (**c**) the thermal shock lives (**d**) substrate temperature diagrams of all coated samples [147].

During the manufacture of FG-TBCs, the quenching of flat particles causes deposition stresses, and cooling generates thermal expansion coefficient mismatch stresses. The thermal stresses have the potential to cause crack expansion in FG-TBCs, which is critical to their performance. Khor et al. [145] showed that the residual stress increases with decreasing coating thickness. The residual stress decreased as the number of graded layers increased for the same size of FG-TBC. Ramaswamy et al. [150] pointed out that the residual stress of three-layer FG-TBCs is 1.8 times higher than that of the conventional 8YSZ coating with the same thickness. On the other hand, FG-TBCs have a longer lifetime than conventional 8YSZ (Figure 24). The thermal stress of FG-TBCs is influenced by the coating-to-substrate thickness ratio, the number of layers, and the cooling rate.

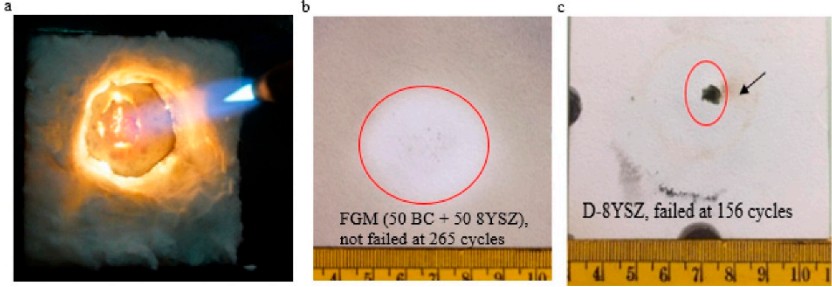

**Figure 24.** Photographs of (**a**) TBC under thermal gradient test at 1200 °C (**b**) FG-TBC not failed at 265 cycles (**c**) traditional 8YSZ cracked at 156 cycles [150].

In summary, the functional gradient coating improves the fracture toughness near the TC/BC interface, which prevents crack extension caused by the greater tip strain energy than the fracture toughness of the material under a thermomechanical load. The reduction of thermal stress in the manufacturing process of an FG-TTBC has a positive effect on its lifetime. Functionally graded TTBCs combined improved the thermal insulation and the lifetime of the coating.

### 3.1.2. TTBCs with Segmented Cracks and Columnar Structure

Columnar-like TTBCs combine the advantages of higher strain tolerance and better crystal structural stability [128,130,151]. Segmentation cracks are cracks that run perpendicular to the coating surface and penetrate at least half of the coating thickness. The columnar structure and segmented TTBCs improve the thermal cycling resistance compared to the traditional lamellar coating due to the increased compliance of the coatings [152,153].

Many researchers have developed segmented or dense vertical crack coating microstructures using the APS process [50,74,131,154,155]. Lu et al. [156] reported that vertical-type cracks developed in TCs are essential for improving the lifetime performance of TTBCs in a high-temperature environment. In addition, the segmentation crack density is closely related to the lifetime of TBCs. Bengtsson et al. [157] reported that substrate temperature and passage thickness are the major parameters determining the segmentation crack density of coatings. Schwingel et al. [74] showed that the thick TBCs containing a high segmentation crack density had an improved thermal shock lifetime. Guo et al. [118] produced three TTBCs with 1.5 mm TCs using APS at different process parameters. The segmentation crack densities of the three TTBCs were 2.7 mm$^{-1}$, 1.5 mm$^{-1}$, and 0.9 mm$^{-1}$, respectively (Figure 25). At 1200 °C, the thermal conductivity of the 2.7 mm$^{-1}$ crack density coating was 1.75 W/m·K, which is 30% higher than the other two because of the heat flow shortcut provided by segmented cracks. The results show that a high segmentation crack density gave rise to improved thermal cycling life of TBCs. The coating with a 2.7 mm$^{-1}$ crack density had a lifetime of more than 1770 cycles at 1238 °C and approximately 300 cycles at 1335 °C (Figure 26). Wang et al. [158] examined how much the segmentation crack density is appropriate for improved TBC thermal shock resistance based on finite element simulations and PS-TTBC thermal shock tests. An increase in the density of splitting cracks within a specific range is followed by a decrease in the elastic modulus and induced coating stress, enhancing the thermal cycle life of the coating (Figure 27). The radial stress distribution of the TTBC based on FEM calculations showed that segmented cracks will reduce the tensile stress at the crack tip, releasing the stress concentration in TTBCs (Figure 28). The stress intensity factor and the energy release rate of cracks fade and then increase with increasing segmentation crack density (Figure 29), so there is an upper limit to the positive effect of the segmentation crack density on the lifetime of TBCs. The calculations showed that segmentation crack density in the range of 2.38–4.76 mm$^{-1}$ are beneficial for improving the thermal shock resistance. However, the effect of the segmentation crack density on the lifetime of TBCs differs according to the thickness, and the segmentation crack density should be investigated in relation to the thickness of TBCs in the future. Tailor et al. [159] reported that PS-TBCs can also implant segmented cracks by post-treatment after deposition with a controlled segmentation crack density. This is an effective way to tailor the appropriate segmentation crack density of TBCs. The high segmentation crack density imparts stronger strain tolerance for TBCs, but there is a partial loss of thermal insulation due to the vertical crack direction in line with the heat flow. Moreover, segmented cracks may be a large passage for oxygen transport from the environment to the interior of TBCs [152].

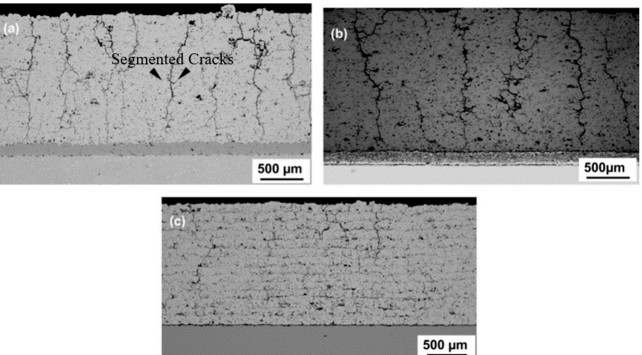

**Figure 25.** SEM micrographs of cross-section of as-sprayed YSZ-TTBCs, the segmentation crack densities of (**a**–**c**) are 2.7 mm$^{-1}$ crack density, 1.5 mm$^{-1}$ crack density and 0.9 mm$^{-1}$ crack density [118].

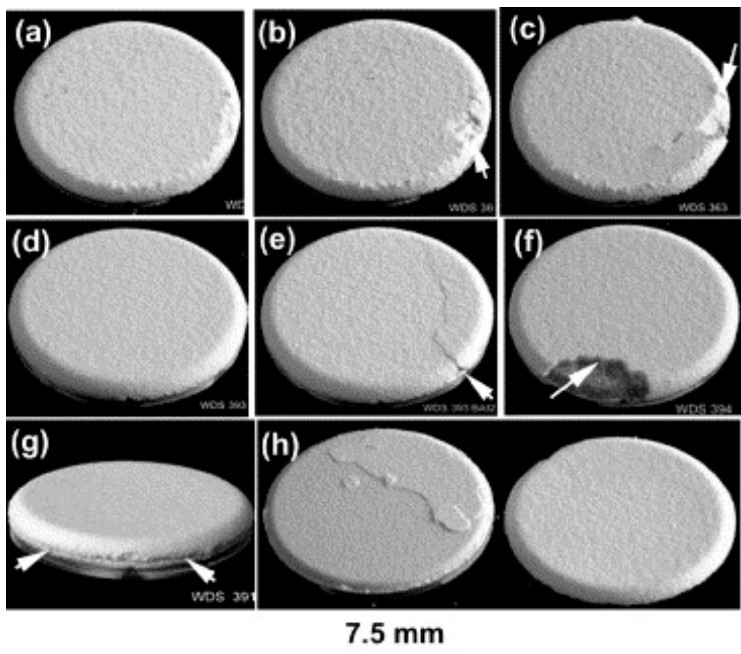

**Figure 26.** Photographs of TBCs showing coating failure modes: (**a–c**) 2.7 mm$^{-1}$ crack density coating cycled to 1238 °C for 1200 cycles and 1770 cycles, and to 1335 °C for 320 cycles, respectively; (**d–f**) 2.7 mm$^{-1}$ crack density coating cycled to 1226 °C for 1650 cycles and 1810 cycles, and to 1317 °C for 174 cycles, respectively; (**g,h**) 0.9 mm$^{-1}$ crack density coating cycled to 1216 °C for 1071 cycles and to 1327 °C for 217 cycles, respectively [118].

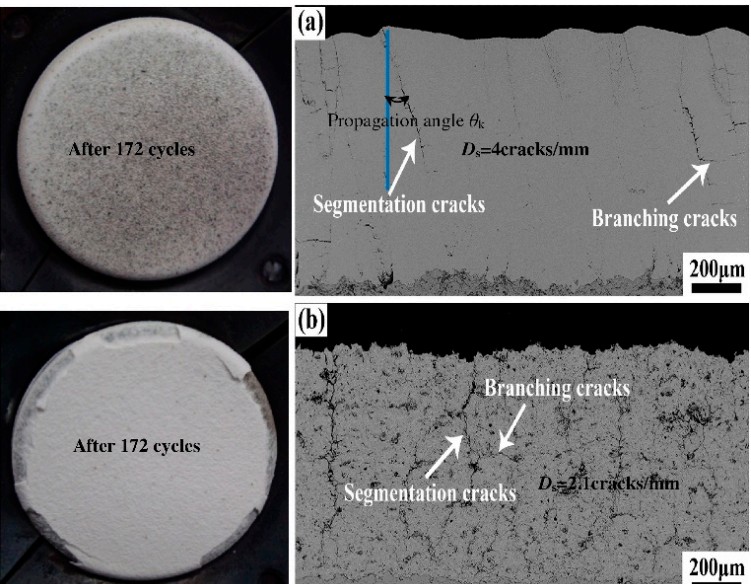

**Figure 27.** Burner rig test results for TTBCs with segmentation crack densities of (**a**) 4/mm and (**b**) 2.1/mm [158].

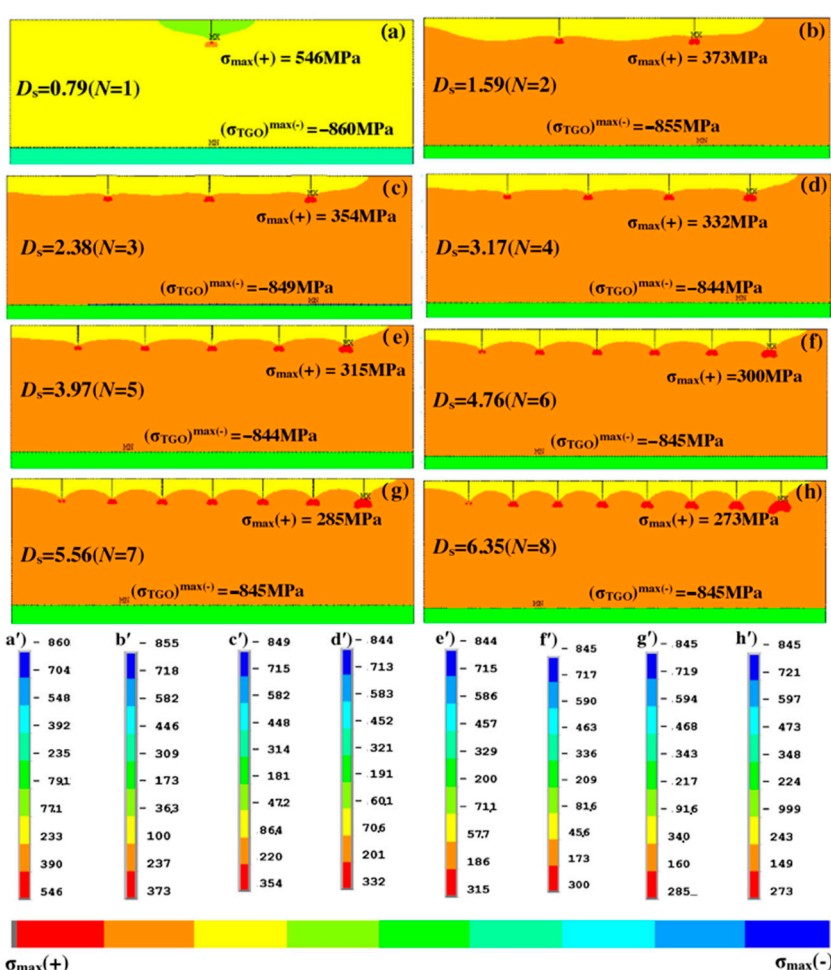

**Figure 28.** Radial stress distribution of TTBCs with different segmentation crack densities [158]. (**a**) 0.79 mm$^{-1}$ crack density; (**b**) 1.59 mm$^{-1}$ crack density; (**c**) 2.38 mm$^{-1}$ crack density; (**d**) 3.17 mm$^{-1}$ crack density; (**e**) 3.97 mm$^{-1}$ crack density; (**f**) 4.76 mm$^{-1}$ crack density; (**g**) 5.56 mm$^{-1}$ crack density; (**h**) 6.35 mm$^{-1}$ crack density, (**a′–h′**) shows the scalar of the contour of the corresponding figures (**a–d**) (unit: MPa).

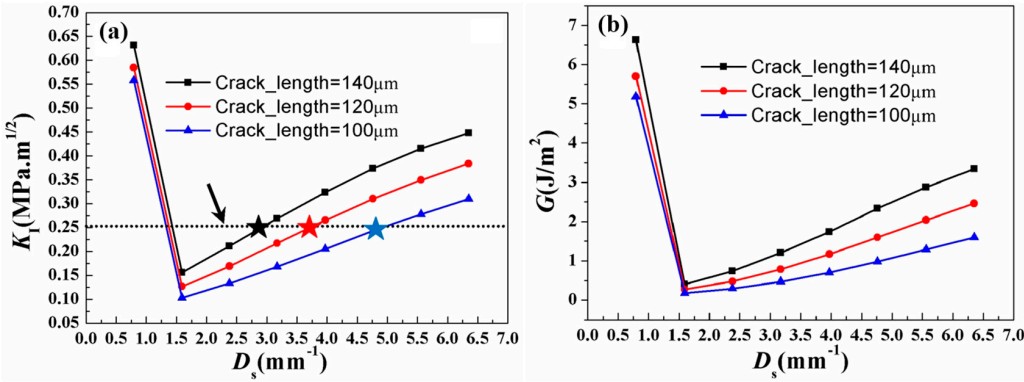

**Figure 29.** (**a**) The plot of stress intensity factor (the intersection point at the right side of each plotted curve can be defined as the upper crack density for the specific TTBCs with improved thermal shock resistance) and (**b**) the energy release rate of the segmentation crack as the function of the segmentation crack density [158].

In a PS-PVD plasma plume diameter of 200−400 mm, the length can be extended to more than two meters, and temperatures to more than 6000 K because of the low working

pressure (50–200 Pa) and high power (~120 kW) plasma gun [160–162]. When spraying with fine-grained powders, the powder can be evaporated at a high nucleation rate to obtain coatings with columnar microstructures [163].

The supersaturation distribution of the plasma jet in the near-substrate region leads to changes in the coating morphologies. The decrease in supersaturation in the edge region of the plasma jet causes a decrease in temperature. This leads to condensation of the gaseous phase and incomplete evaporation of the coating powder in the external plasma jet, causing the coating to exhibit a different structure in the radial direction of the plasma jet [164], from the center to the edge of the plasma jet, where the coating structure will change from EB-PVD-like columnar to quasi-columnar, or even to dense (Figure 30) [165]. Adjusting the plasma gas composition and feed rate of the PS-PVD process can allow the formation of different microstructure TBCs, including lamellar coatings, EB-PVD-like columnar coatings, quasi-columnar coatings, and composite structured coatings [166–170].

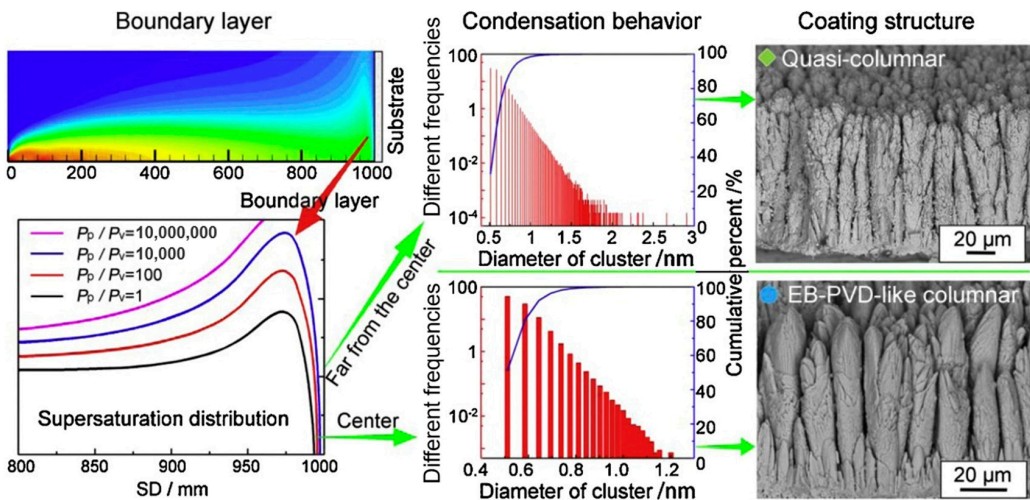

**Figure 30.** Effect of condensation behavior on the coating structure [164,165].

Gao et al. [171] manufactured a YSZ quasi-columnar-structured coating using PS-PVD and examined the thermal conductivity and thermal cycling behavior of the coating. The quasi-columnar-structured YSZ coating had a lifetime of 2000 cycles in the thermal cycling test at 1200 °C while exhibiting relatively low thermal conductivity ~1.15 W/m·K at 1200 °C. The columnar-structured coating prepared by PS-PVD was formed by stacking small columns. Thus, the interface between the small columns may enhance the thermal insulation of the coating [172]. Qiu et al. [173] prepared quasi-columnar-structured YSZ and GZO coatings by PS-PVD and developed a three-dimensional, quasi-columnar-structured coating model. The coating exhibited a typical quasi-columnar structure, where the columns are composed of many small island-like columns stacked on top of each other. Hence, there are many transverse interfaces and pores in the columnar structure (Figure 31), which can enhance the thermal insulation of the coating while increasing the strain tolerance of the coating. The model represents a large heat flux at the interface created by the stack of island-like columns (Figure 32). More interfaces indicate a stronger barrier to the heat flow because, in a steady-state heat transfer, a larger heat flux means a smaller effective contact area. However, the unique columnar PS-PVD-TBCs are expected to have a low resistance to CMAS and solid particle erosion because of the large amount of open porosity.

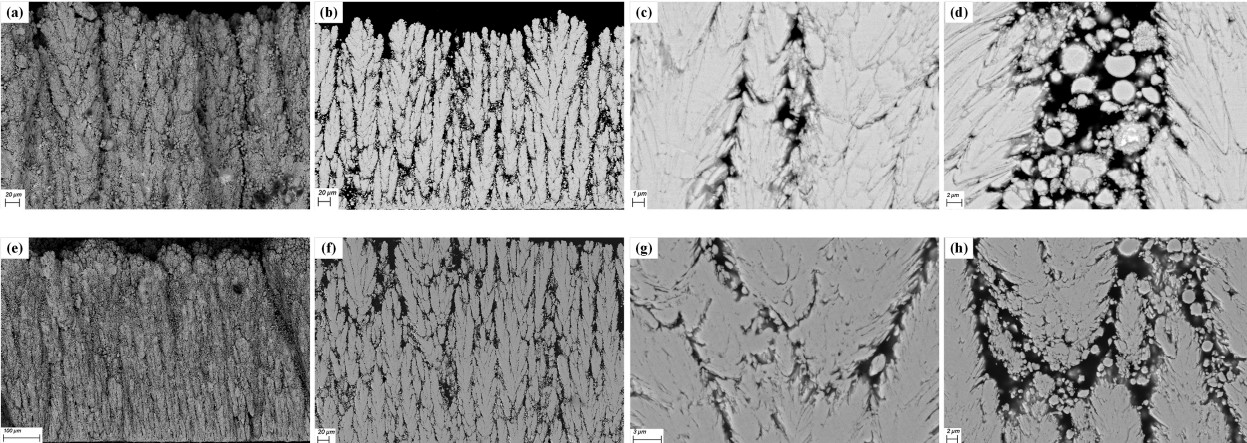

**Figure 31.** The SEM micrograph of the PS-PVD coatings, (**a**) fractured cross-section of the YSZ coating (**b**) polished cross-section of the YSZ coating (**c**,**d**) voids and interfaces between small columns of the YSZ coating (**e**) fractured cross-section of the GZO coating (**f**) polished cross-section of the GZO coating (**g**,**h**) voids and interfaces between small columns of the GZO coating [173].

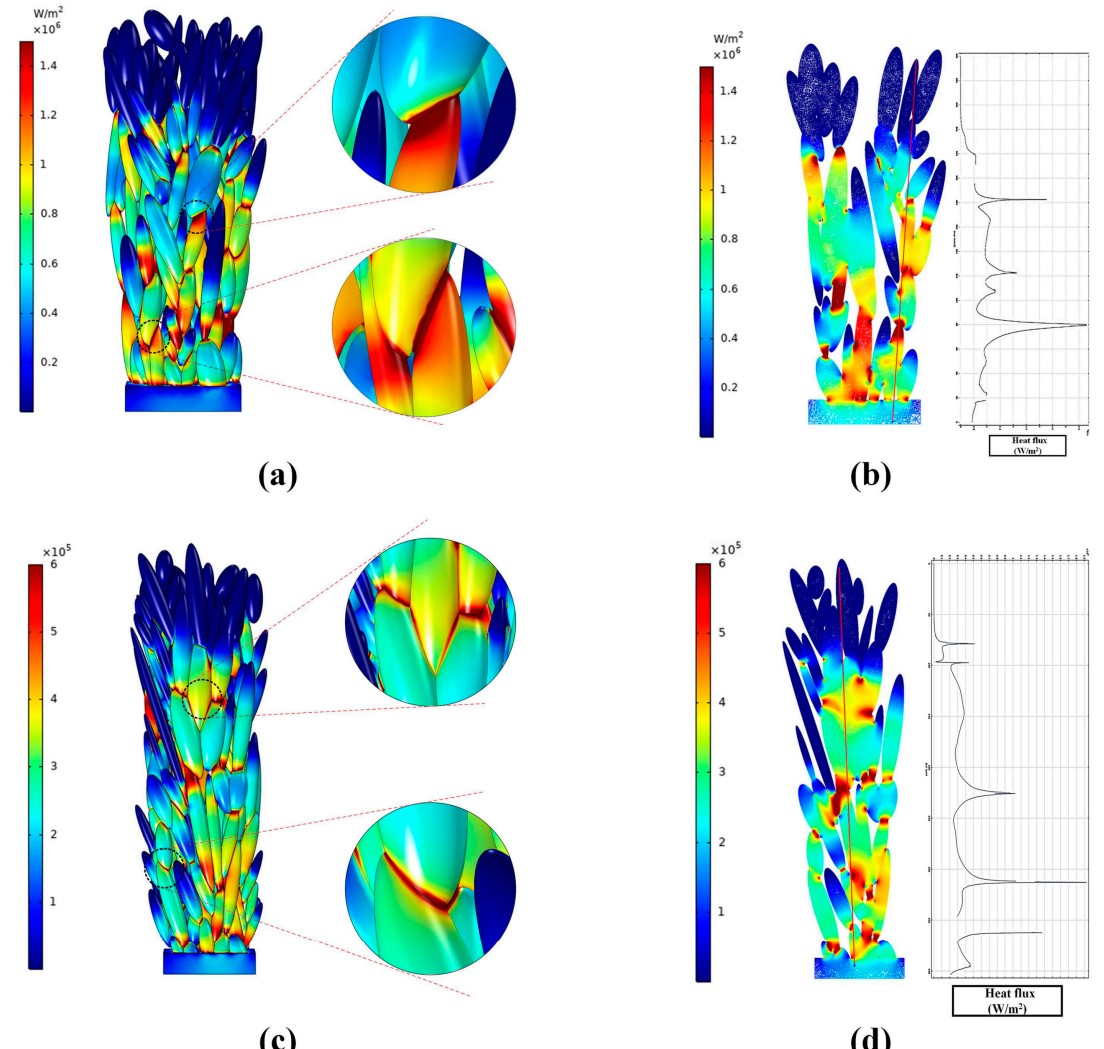

**Figure 32.** The YSZ coating model: (**a**) heat flux field (**b**) cross-section of heat flux field, the GZO coating model (**c**) heat flux field (**d**) cross-section of heat flux field [173].

In short, PS-PVD-TBCs possess a columnar structure containing many pores and interfaces, with the advantages of both APS lamellar and EB-PVD columnar structures. However, the unique microstructure produces conditions for interactions with liquid CMAS deposits or solid eroded particles. Therefore, improving erosion resistance is the key to the future development of PS-PVD-TBCs.

Some experimental studies showed that SPS TBCs exhibit unique microstructures of a high segmentation crack density with relatively high porosity levels simultaneously, which contribute to their superior thermal cycling durability and low thermal conductivity [174]. These have the potential for fabricating this type of high-strain-tolerated TTBCs.

The SPS process uses liquid feedstock, allowing control of the microstructure of the spray on the nanoscale [175–177]. When the suspension feedstock is injected into the plasma jet, the feedstock is broken into many small droplets because of the large velocity difference. The droplets fly with the plasma gas flow, and when the gas flow encounters a rough surface (substrate and the surface deposited first), it is deflected. Small droplets move with it and impinge obliquely on the protrusions of the surface [178,179]. Eventually, the stacked droplets gradually form a columnar structure due to the shadowing effect [180].

Zhao et al. [181] produced YSZ-TTBCs with segmented cracks by both SPS and APS processes (Figure 33). The segmentation crack densities of SPS-TTBCs and APS-TTBCs were 4 and 2.5 mm$^{-1}$, respectively. The results showed that the thermal shock resistance of the SPS-TTBC was improved approximately twofold compared with that of the APS-TTBC (Figure 34), which may be due to the increased segmentation crack density and superior segment structure of SPS-TTBCs. In addition, the failure of the SPS-TTBC is ultimately connected to the thermal stress and the severe oxidation of BCs. In conclusion, segmented cracks and columnar microstructure play important roles in controlling the service life of TTBCs.

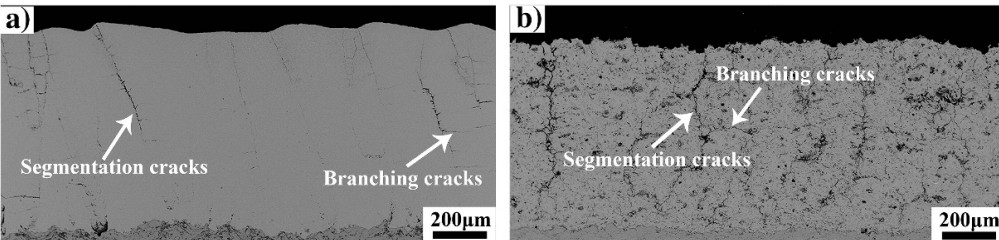

**Figure 33.** Cross-sectional microstructures of as-sprayed TTBCs: (**a**) SPS (**b**) APS coating [181].

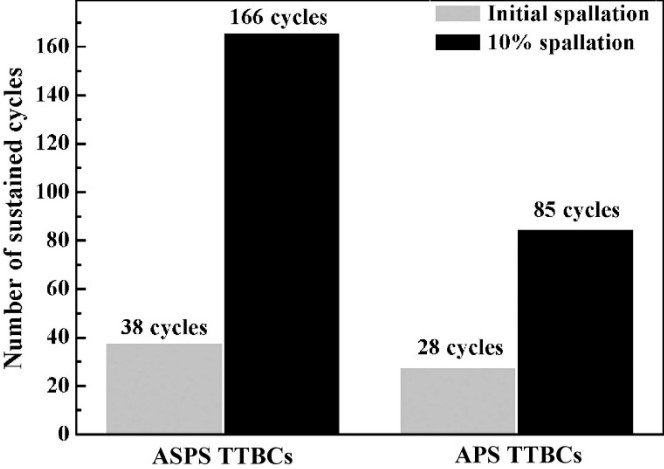

**Figure 34.** Number of sustained cycles at 1100 °C for SPS coating and APS coating [181].

### 3.2. Advanced Materials Build Double Ceramic Layers

High thermal expansion coefficients, ultralow thermal conductivity, reasonable mechanical properties, and outstanding damage tolerance are essential for promising TBC materials [79,182–186]. YSZ is a classical and widely used TC material because of its low thermal conductivity and good mechanical compatibility with the matrix. However, the damage tolerance and phase stability of YSZ are lost when operated at temperatures above 1200 °C. In addition, as turbine intake temperatures increase, there is an increasing need for materials for TBCs with smaller thermal conductivity and highly stable phases at elevated temperatures. Recently, many researchers focused on developing rare-earth oxide materials [4,187–197]. Table 1 lists the potential TC materials and their corresponding properties.

**Table 1.** Thermal physical properties of potential TC materials.

| Material | Thermal Conductivity /$(W \cdot m^{-1} \cdot K^{-1})$ | Thermal Expansion Coefficient/$(10^{-6} K^{-1})$ | Elastic Modulus/ (GPa) | Fracture Toughness/ (MPa m$^{1/2}$) |
|---|---|---|---|---|
| YSZ [4,187] | 2.20 (1000 °C) | 10.70 (1000 °C) | $210 \pm 10$ | $3 \pm 0.5$ |
| | | Rare-earth zirconates | | |
| $La_2Zr_2O_7$ [68,182,189] | 1.56 (1000 °C) | 9.10 (1000 °C) | 186 | $2.1 \pm 0.1$ |
| $Gd_2Zr_2O_7$ [188–190] | 1.60 (700 °C) | 11.60 (1000 °C) | 205 | $2.2 \pm 0.2$ |
| $Dy_2Zr_2O_7$ [191] | 1.34 (800 °C) | 10.80 (1000 °C) | | |
| $Yb_2Zr_2O_7$ [192] | 1.36 (800 °C) | 11.90 (1000 °C) | | $2.2 \pm 0.1$ |
| | | Rare-earth tantalates | | |
| $YTaO_4$ [193] | 1.5 | | 140 | 5 |
| $NdTaO_4$ [194] | 1.41 (800 °C) | | 178 | |
| $GdTaO_4$ [195] | 1.70 (800 °C) | | 154 | |
| $YbTaO_4$ [193] | 1.65–3 | 7.6 (1200 °C) | 122 | |
| $EuTaO_4$ [195] | 1.26 (900 °C) | | 175 | |
| | | Rare-earth niobates | | |
| $NdNbO_4$ [196] | 2.0 (1000 °C) | 11.80 (1000 °C) | 108 | |
| $SmNbO_4$ [196] | 2.05 (1000 °C) | 10.86 (1000 °C) | 120 | |
| $GdNbO_4$ [196] | 1.66 (1000 °C) | 10.30 (1000 °C) | 131 | |
| | | Other oxides | | |
| $CaZrO_3$ [197] | 0.73 (600 °C) | 9.00 (1000 °C) | | |
| $SrZrO_3$ [198] | 2.08 (1000 °C) | 10.90 (1000 °C) | $170 \pm 4$ | $1.5 \pm 0.1$ |
| $BaZrO_3$ [199] | 3.42 (1000 °C) | 7.90 (1000 °C) | $181 \pm 11$ | |
| $YAlO_3$ [200] | 1.61 (min) | | 318 | |
| $YbAlO_3$ [201] | 1.15 (min) | 9.62 | 257 | |
| $LaPO_4$ [202] | 1.30 (1000 °C) | 10.50 (1000 °C) | 131 | $1.1 \pm 0.2$ |
| $CePO_4$ [202] | 1.35 | 9.9 | | $1.8 \pm 0.1$ |
| $NdPO_4$ [202] | 1.59 | 9.8 | | $2 \pm 0.2$ |
| $LaMgAl_{11}O_{19}$ [203] | 1.95 (1000 °C) | 10.95 (20~1000 °C) | $130 \pm 11$ | $4.6 \pm 0.5$ |
| $LaTi_2Al_9O_{19}$ [204] | 1.30 (1000 °C) | 11.2 (1000 °C) | $240 \pm 13$ | 1.9–2.5 |

#### 3.2.1. Rare-Earth Zirconates

The physical properties of rare-earth zirconates ($RE_2Zr_2O_7$) are closely related to their unique structure. Rare-earth zirconates usually have either a pyrochlore structure (Figure 35a) or a fluorite structure. Both structures are face-centered cubic space lattices, with the main difference being the ordered distribution of oxygen vacancies.

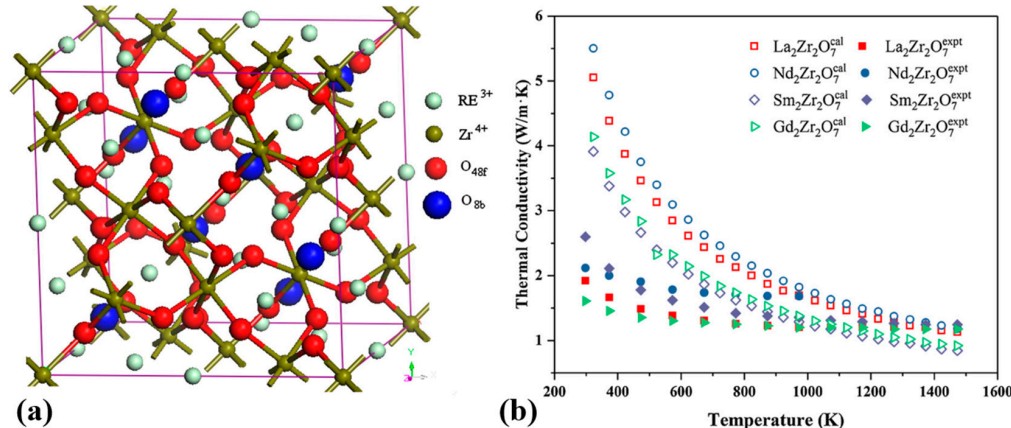

**Figure 35.** (**a**) Crystal structure (**b**) calculated and experimental temperature-dependent thermal conductivities of the rare-earth zirconate [189,205,206].

The pyrochlore structured $RE_2Zr_2O_6O'$ belongs to the Fd3m (227) space group. A complete cell contains eight molecular units of $RE_2Zr_2O_7$ with four unequal crystallographic atomic positions of L, Zr, O, and O'. There are three oxygen ion lattice positions: 8b, 48f, and 8a, where the space positions of 8b and 48f are occupied by O' and O, respectively; the oxygen vacancy occupies the space position of 8a in the tetrahedron formed by four $Zr^{4+}$. The fluorite structure belongs to the Fm3m (227) space group. The cation in the fluorite structure has only one crystallographic position, and the central position of the cation is occupied by the oxygen ions. The coordination number of the cation is seven, and there are 1/8 intrinsic oxygen vacancies with randomly distributed positions in the crystal structure.

However, pyrochlore and fluorite structures contain intrinsic oxygen vacancies in each crystal structure unit. The high concentration of oxygen vacancies and the larger mass of rare-earth atoms in the crystal cell enhance the phonon scattering effect, decreasing the mean-free range. Therefore, rare-earth zirconates have a low thermal conductivity. The increase in oxygen vacancy concentration increases the asymmetry of the potential well, which favors the increase in CTE. The CTE of $RE_2Zr_2O_7$ increases as the radius of the rare-earth ions decreases. In addition, doping can introduce defects to relax the lattice and reduce the lattice energy that increases the CTE.

Feng et al. [189,205] used a quasi-harmonic approximation combined with first-nature principle calculations to investigate the thermal conductivity of $RE_2Zr_2O_7$. Figure 35b presents the calculated and experimental temperature-dependent thermal conductivity of $RE_2Zr_2O_7$. The agreement between the calculated and experimental values was good at high temperatures. For $RE_2Zr_2O_7$, the predicted value at 1273 K was approximately 0.98–1.37 W/m·K, which is comparable to the experimental value of approximately 1.2 W/m·K. However, the bond strength of the rare-earth zirconate coating was not as strong as that of the conventional YSZ coating. The selection of appropriate powder particle size and spraying parameters can improve the bond strength [108,109].

3.2.2. Rare-Earth Tantalates

Recently, there has been a promising development in the study of rare-earth tantalates ($RETaO_4$) being applied as thermal barrier coating materials. Rare-earth tantalates are mostly monoclinic phases at room temperature and can be used at more than 1600 °C. The thermal conductivity tends to decrease with increasing temperature in a specific temperature range.

Clarke and Levi examined yttrium tantalate ($YTaO_4$), which is ferroelastic [207,208]. They showed that the stable phases of $YTaO_4$ at high and room temperatures are tetragonal (t) and monoclinic (m), respectively. The phase transition temperature is approximately 1430 °C. The t–m ferroelastic transformation occurs during the cooling process [209,210]. Unlike the phase transition process of the YSZ, the high-temperature ferroelastic phase tran-

sition is secondary. The change in cell volume and temperature after the phase transition is continuous and no volume change occurs [211].

In addition to the m-phase, RETaO$_4$ belongs to the m' structure when fabricated using elevated temperature solid-state reactions (Figure 36) and has good thermal stability and thermal protection properties. According to Chen et al. [193], m' RETaO$_4$ exhibits outstanding lattice stability with structural stability up to 1500 °C. However, the shorter bond length and stronger bond strength result in a lower CTE and higher elastic modulus.

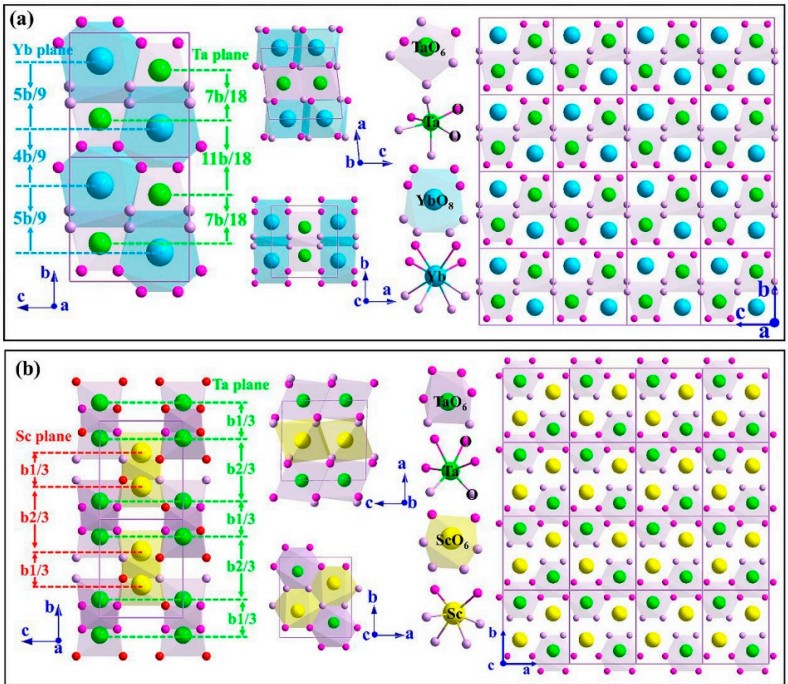

**Figure 36.** Characteristics of crystal structures in m' phase RETaO$_4$ (RE = Yb, Lu, Sc) ceramics: (**a**) YbTaO$_4$ (**b**) ScTaO$_4$ [193].

Figure 37 shows the microstructures of RETaO$_4$. Granular RETaO$_4$ was observed in the microstructure with a grain size of approximately 2–14 μm, and the domain structure was clearly visible. Ferroelastic domains can deflect cracks generated during material cracking and absorb the fracture energy. Thus, fracture toughness is improved, resulting in better thermal and mechanical properties of RETaO$_4$. The deposition and thermal exposure behavior of tantalate rare-earth coatings should be investigated further.

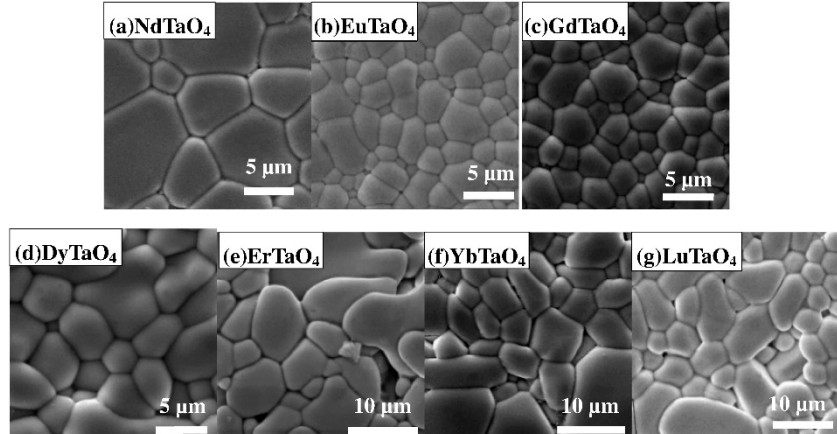

**Figure 37.** SEM photographs of RETaO$_4$ (RE = Nd, Eu, Gd, Dy, Er, Yb, Lu) ceramics [194].

### 3.2.3. Rare-Earth Niobates

Rare-earth niobate is a new ceramic material with an integrated structure and function that has attracted considerable attention. The basic crystal structures are pyrochlore with a defective fluorite structure ($RE_3NbO_7$) and perovskite structure ($RENbO_4$) (Figure 38a). Perovskite rare-earth niobate $RENbO_4$ ceramics have a similar crystal structure to $RETaO_4$ ceramics and have the same potential as a new TBC material. $RENbO_4$ exhibits a lower thermal conductivity than YSZ at > 200 °C (Figure 38b), with a range of 1.80–2.26 W/m·K at 1000 °C. The low thermal conductivity of $RENbO_4$ is due to the large chemical inhomogeneity in terms of the charge difference between cations and a fluctuating bonding length. Rare-earth niobate has a ferroelastic structure (Figure 39), which allows for increased fracture toughness.

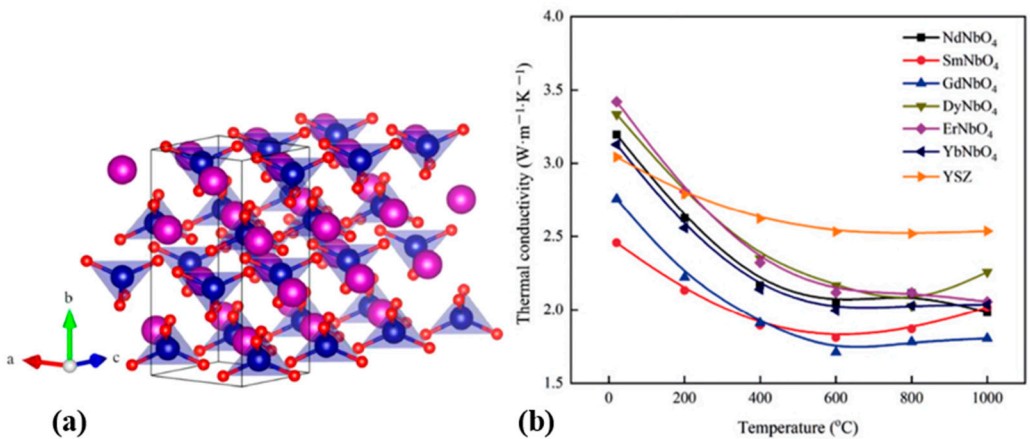

**Figure 38.** (**a**) Crystal structure (**b**) thermal conductivity of $RENbO_4$ ceramic [196].

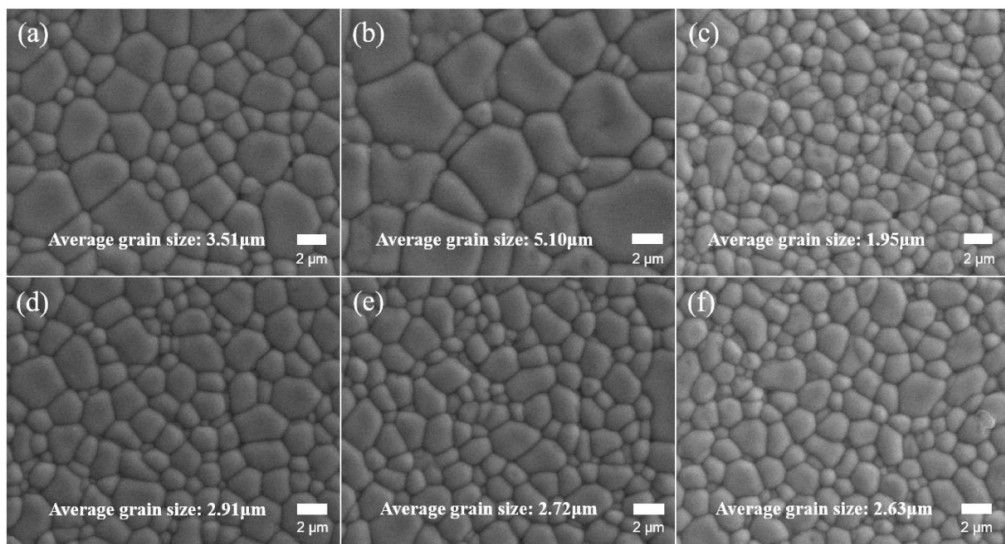

**Figure 39.** The microstructure of $RENbO_4$ ceramics sintered at 1600 °C for 10 h: (**a**) $NdNbO_4$ (**b**) $SmNbO_4$ (**c**) $GdNbO_4$ (**d**) $DyNbO_4$ (**e**) $ErNbO_4$ (**f**) $YbNbO_4$ [196].

### 3.2.4. Other Materials

Perovskite-oxides ($ABO_3$, where A = Ba, Sr, and B = Zr, Ti, Hf) can accommodate a wide range of ions in the solid solution, including those with large atomic masses, and are considered potential ceramic materials for TBCs because of their excellent mechanical properties. $CaZrO_3$ has low thermal conductivity and cost, but the thermal expansion and melting point coefficients are lower than YSZ, which can be applied to some lower-temperature environments [197,212–214]. $RAAl_{11}O_{19}$ is an alumina-based coating material

with a low sintering rate and good thermal stability, which can be used in environments above 1200 °C [215]. However, it is unsuitable for the EB-PVD method preparation due to the low deposition efficiency and amorphous body [215]. In summary, poor fracture toughness and low CTE are the main reasons limiting the use of advanced materials for TC (Figure 40). Several studies have designed different structures with different materials and tried to overcome these problems.

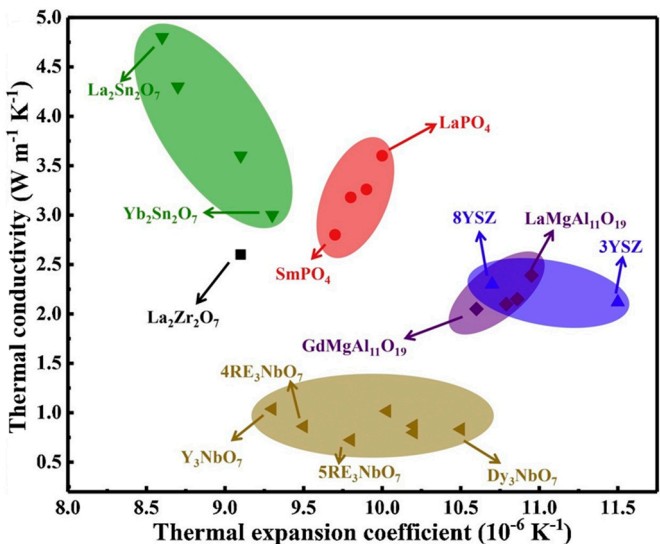

**Figure 40.** Thermal conductivity versus thermal expansion coefficient [216].

### 3.2.5. YSZ-Based Double Ceramic Layer

A double ceramic layer (DCL) structure is an effective means to overcome the poor fracture toughness of ceramic materials for advanced TBCs. The DCL structure is based on depositing a layer of advanced ceramic material on YSZ. A DCL is a better solution to improve the thermal insulation ability and bond strength of the coating simultaneously. The bottom layer of YSZ enhances the fracture toughness of the coating and provides a transition adjustment for the thermal expansion mismatch between the advanced material and substrate. The advanced material used as the top layer reduces the thermal conductivity of the coating and enhances the thermal insulation performance of the coating.

Cheng et al. [217,218] prepared $La_2Zr_2O_7$ (LZO)/YSZ DCL TBCs with different layer thickness ratios and different elastic moduli by APS (Figure 41). Figure 41a presents a diagram of five TBC structures with different layer thickness ratios having the same thermal insulation effect, where 67 μm LZO and 100 μm YSZ have the same thermal insulation effect. Thus, advanced ceramic materials help reduce the thickness of ceramic layers while maintaining the same thermal insulation performance. Figure 42a shows the cross-sectional microstructures of five TBCs, where the coatings of two materials have similar microstructures because of the same preparation method. Under the thermal gradient cycling test, 4Y/1L and 3Y/2L exhibited higher lifetimes than the conventional YSZ coating (Figure 43a). As the percentage of LZO increased, the coating failure shifted from the TC/BC interface to the LZO/YSZ interface and finally to within the LZO layer because of the lower CTE and poorer fracture toughness of LZO. Subsequently, the combination of 3Y/2L was optimized, and LZO layers with different elastic moduli were prepared (Figure 41b). LZO with different elastic moduli was achieved by varying the spraying distance to prepare different porosity structures (Figure 42c−e). By further optimization of the microstructure, the TBC lifetimes of the 3Y/2L combination were all higher than those of the conventional YSZ. The thermal gradient cycle lifetimes were, in descending order, 3 Low LZO/2Y > 3 Middle LZO/2Y > 3 High LZO/2Y > 5YSZ (Figure 43b). Chen et al. [219] examined the thermal cycling failure of $LaMgAl_{11}O_{19}$ (LMA)/YSZ DCL TBC with YSZ and LMA monolayer systems. The results showed that the LMA/YSZ DCL TBCs, which overcame the thermal mismatch between the LMA TC and substrate, have better strain

tolerance and thermal cycling life than the single-layer YSZ and LMA coatings. Therefore, DCL TBCs can improve the thermal insulation while enhancing the lifetime.

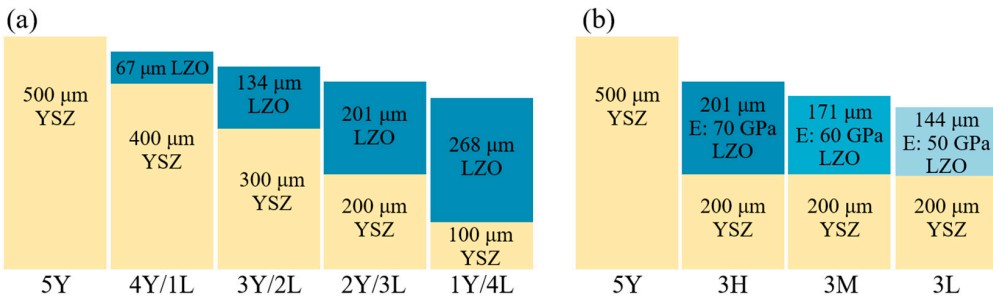

**Figure 41.** Schematic diagram of DCL structure: (**a**) equivalent thermal insulation with different thickness ratio (**b**) equivalent thermal insulation with different modulus [217,218].

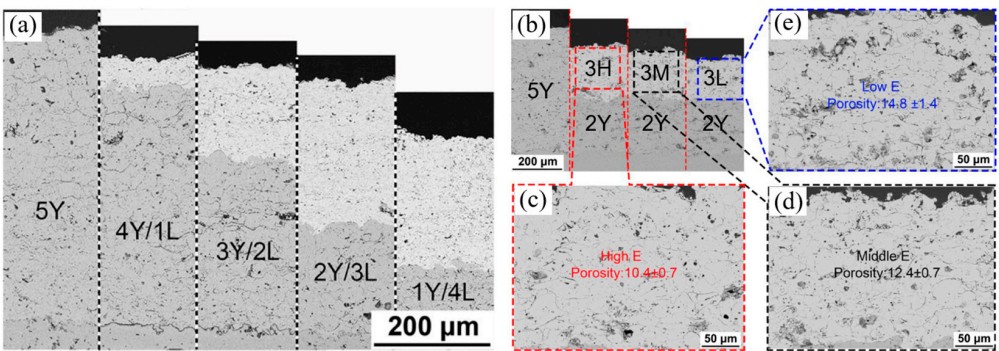

**Figure 42.** Cross-sectional SEM images of DCL TBCs: (**a**) equivalent thermal insulation with different thickness ratio (**b**–**e**) equivalent thermal insulation with different modulus [217,218].

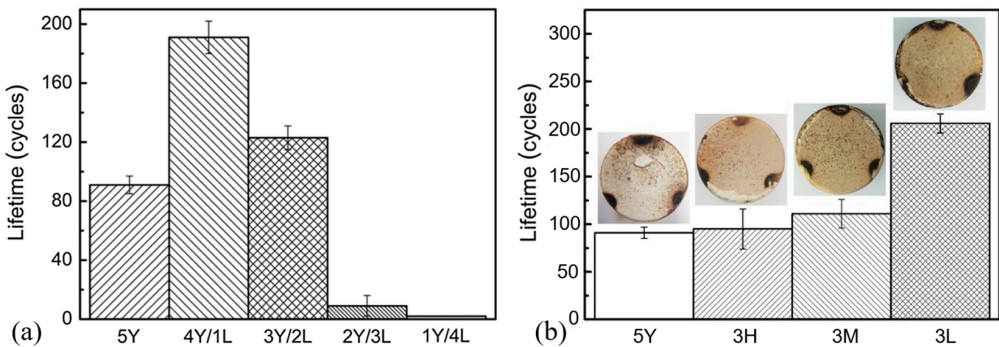

**Figure 43.** Lifetime of DCL TBCs under gradient thermal cyclic tests at 1300 °C TC surface and 940 °C backside surface: (**a**) equivalent thermal insulation with different thickness ratio (**b**) equivalent thermal insulation with different modulus [217,218].

### 3.3. Nano- and Composite-Structured Coatings

3.3.1. Nanostructured Coatings

The mechanical properties of materials can be enhanced significantly when the grain size of a material is transformed from the microscale to the nanoscale [220–222]. The lifetime of nanostructured YSZ TBCs has been extended by 50–100% compared to conventional PS YSZ TBCs [223,224]. In addition, the resistance to sintering and oxidation is enhanced significantly [225], and the degradation of the thermal insulation properties is less than 50% [226,227]. Therefore, nanostructured coatings have become a potential next step in developing TBCs.

PS should be carried out by choosing the appropriate spraying parameters [228,229] to achieve a particle state where complete and partial melting coexist and preserve the nanoscale of the raw powder [230]. Thus, the nanostructured PS-TBCs have a bimodal structure, which is manifested as a conventional lamellar structure with nanozones contained in it [225,231]. In nanostructured TBCs, the lamellar zones are relatively dense with anisotropic structures, whereas the nanozones have a higher surface area with an isotropic loose structure [232]. Therefore, the nanozones have a different sintering behavior from the lamellar zone.

Figure 44 shows the microstructural evolution of conventional PS-TBCs and nanostructured TBCs during thermal exposure [233]. After thermal exposure, the initial lamellar multi-scale pore structure of conventional TBCs subsided, and the coating became much denser due to sintering. However, for nanostructured TBCs, the porous nanostructured agglomerates within the nanozones have a higher driving force for sintering-induced densification than the lamellar zones. Therefore, rapid densification of the nanozones led to the formation of coarse voids.

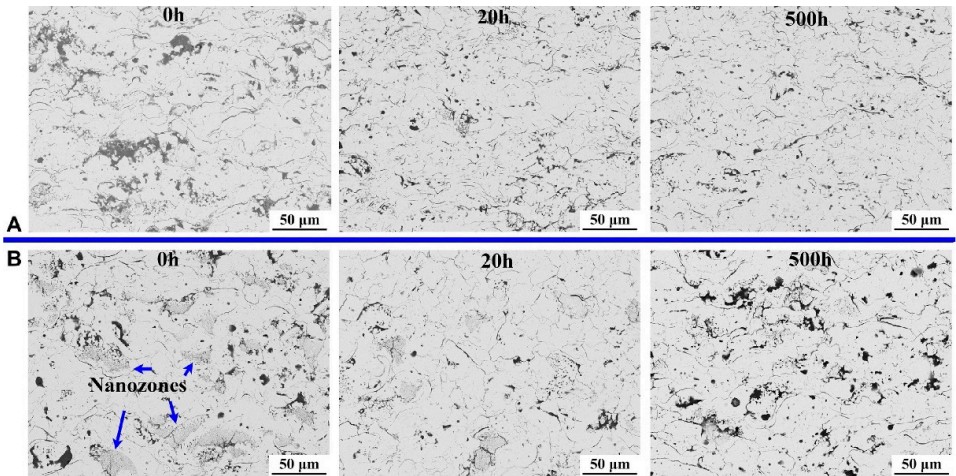

**Figure 44.** Global structural evolution during thermal exposure: (**A**) conventional TBCs (**B**) nanostructured TBCs [233].

Figure 45 shows the changes in the elastic modulus and thermal conductivity as a function of thermal exposure [233]. The changes in both are nonlinear throughout the heat exposure phase. This is consistent with that mentioned in Section 2.3. However, for nanostructured TBCs, the degree of the increase in performance is smaller than that for conventional TBCs. The main reason for this is the contribution of the newly formed coarse pores to the thermal conductivity and strain tolerance. A previous study reported that nanostructured TBCs could achieve longer lifetimes during thermal shock tests [223].

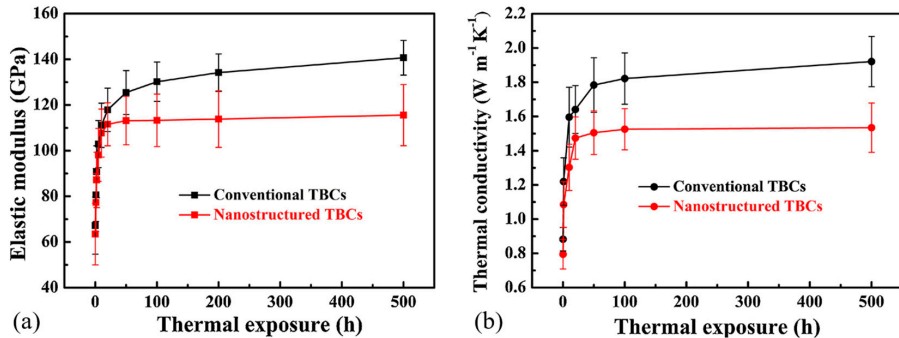

**Figure 45.** Changes in (**a**) elastic modulus (**b**) thermal conductivity as a function of thermal exposure duration [233].

Briefly, nanostructured TBCs are used to maintain the strain tolerance of a coating by comparing the nanozones. The original 2D pores in the lamellar zone still subside. Further efforts should be made to control the newly formed pore orientation, which contributes to the thermal and mechanical properties of the coatings.

### 3.3.2. Composite Structured Coatings

Composite structured TBCs are similar to nanostructured TBCs because both have a bimodal structure, and the spontaneous formation of pores occurs during thermal exposure. However, composite structures can be tailored to a newly formed pore orientation to improve strain tolerance and thermal insulation.

Li et al. [234] designed three TBCs (Figure 46) by tailoring the morphology of the inclusions to obtain a preferred orientation of the insulating pores, namely, conventional TBCs (Mono-TBCs), composite TBCs with isotropic nano-inclusions (composite-TBC-1), and composite TBCs with anisotropic nano-inclusions (composite-TBC-2). Composite-TBC-1 is nanostructured TBCs and composite-TBC-2 is formed by stacking alternately sprayed dense splits and nano-powder heaps with 2D morphology that has a large aspect ratio from the cross-section.

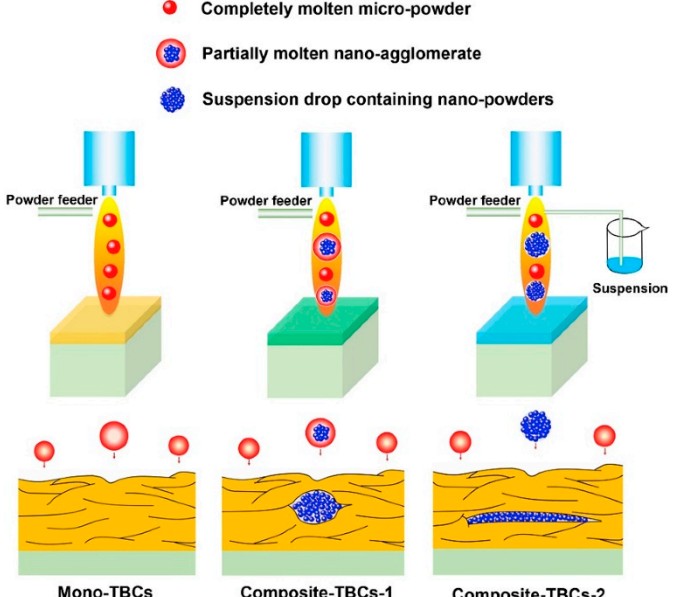

**Figure 46.** Schematic diagram of the three kinds of TBCs: (i) mono-layered TBCs without nano-inclusions (mono-TBCs); (ii) composited TBCs with isotropic nano-inclusions (composite-TBCs-1); (iii) composited TBCs with anisotropic nano-inclusions (composite-TBCs-2) [234].

Figure 47 shows the evolution of the cross-sectional microstructures of the three TBCs during thermal exposure [234]. All TBCs underwent severe sintering, particularly in conventional TBCs, where the 2D pores almost disappeared. These nanostructures form coarse pores after thermal exposure. The difference is that the newly formed pores in composite TBCs-2 have a perpendicular orientation to the heat flow and have 2D pore characteristics. The 2D pores play a significant role in preventing heat flux. The 2D pore density of composite-TBC-2 was consistently higher than that of composite-TBC-1 during heat exposure. Thus, the thermal insulation performance was better (Figure 48).

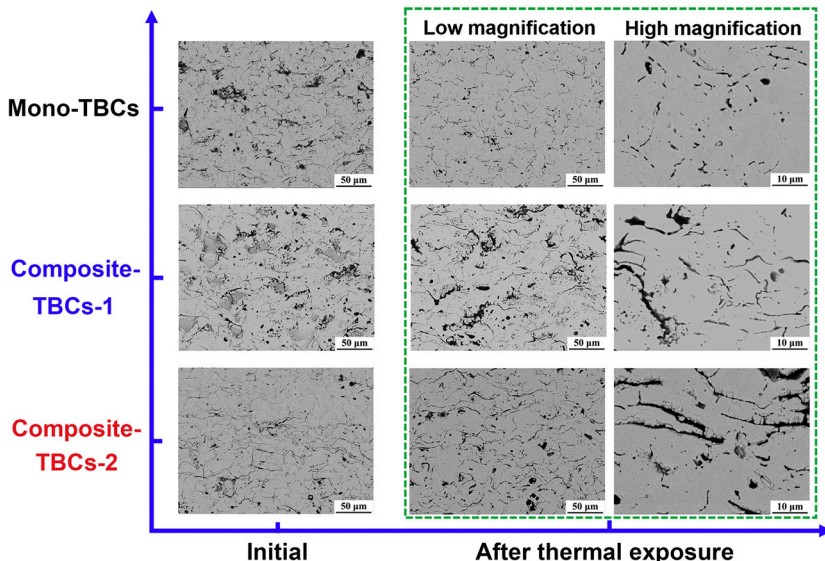

**Figure 47.** Structural evolution of the three kinds of TBCs during thermal exposure [234].

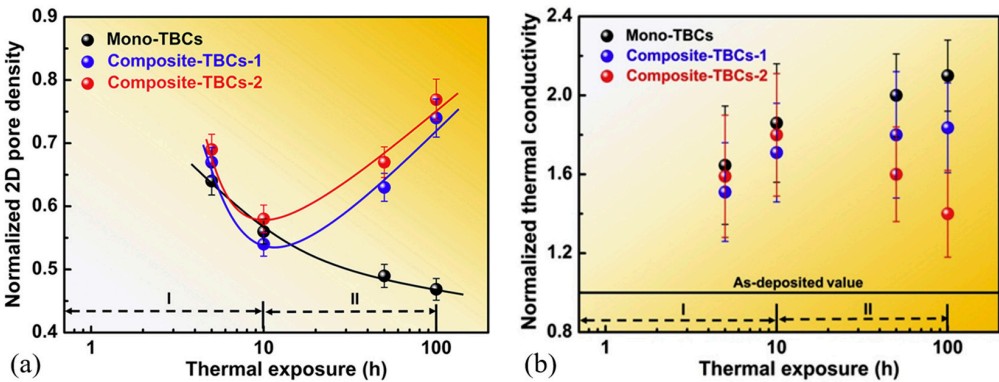

**Figure 48.** Changes in (**a**) 2D pore density (**b**) normalized thermal conductivity during thermal exposure [234].

In conclusion, for the application of composite TBCs, a bimodal structural coating will reduce the increase in thermal conductivity and elastic modulus considerably when exposed to high temperatures by counteracting the densification effect through differential sintering. However, the newly formed pores may reduce the cohesion of the coating. Hence, further investigation is needed.

## 4. Outlook

Thermal barrier coating technology is an indispensable technology for manufacturing high-performance engines in the future. In recent years, researchers have made significant progress in the research of thermal barrier coating preparation technology, but there are still problems in the contradiction between coating performance and equipment cost and process cost, as well as the limitation of application scope. Future research should focus on the following areas: (i) improved PS-TBC deposition process and post-treatment process to achieve porous TBC structures with segmented cracks, and enhanced thermal insulation and strain tolerance of the coating to increase the service life of TBCs; (ii) diversified composite TBC preparation processes, such as plasma enhanced chemical vapor deposition (PE-CVD) and laser CVD, to improve production efficiency and reduce process costs; and (iii) exploring new rare-earth, doped-ceramic material systems and nanoscale ceramic material sizes to improve thermal insulation and bonding of coatings.

## 5. Conclusions

TBCs provide essential thermal protection for gas turbines, and higher thermal insulation and durability of TBCs are required to meet the development of efficient industrial levels. Therefore, this paper reviewed the development and synergistic design of advanced materials and coating structures for TBCs based on a comprehensive understanding of the factors influencing coating thermal insulation and the mechanisms of heat transfer and degradation. The main conclusions are as follows.

(i) The coating thickness intuitively affects the thermal insulation; the TBC thickness increased by 0.2 mm, which results in a decrease in the maximum blade temperature of approximately 40 K. However, it also leads to large thermal gradients and high elastic strain energy, which can drive cracks, leading to coating delamination. Functional gradients and TTBCs with segmented cracks can solve this problem. Functional gradient TTBCs improve the fracture toughness near the TC/BC interface. TTBCs with segmented cracks increase the compliance of the coating, and segmented crack densities between 2.38 and 4.76 mm$^{-1}$ contribute to the thermal shock resistance of the coating.

(ii) Advanced TBC materials, such as rare-earth zirconates, rare-earth tantalates, and rare-earth niobates, have low thermal conductivity. The rare-earth zirconates produce a YSZ-based DCL structure required for their application because of their low fracture toughness and CTE drawbacks. Choosing the correct thickness ratio in the DCL can increase the coating lifetime up to two times that of conventional coatings. Rare-earth tantalates and rare-earth niobates have good fracture toughness owing to their ferroelastic structure, giving them high potential in next-generation TBCs.

(iii) The porosity and aspect ratio significantly affect the thermal insulation of coatings. The decrease in porosity and aspect ratio due to sintering during thermal exposure are the main reasons for the degradation of thermal insulation. Nano- and composite-structured TBCs have unique appeal to improve the resistance of TBCs to sintering. Both are bimodal structures that resist densification by differential sintering to form new pores during thermal exposure. In composite structured TBCs, the thermal insulation of the coating is increased significantly due to the adjustable orientation of the newly formed pores.

**Author Contributions:** Conceptualization, L.W., J.S., H.D. and J.Y.; Data curation, L.W. and J.S.; Formal analysis, L.W., J.S. and H.D.; Investigation, J.S.; Methodology, L.W. and H.D.; Resources, L.W. and J.S.; Software, J.S. and J.Y.; Validation, H.D.; Visualization, H.D. and J.Y.; Writing–original draft, L.W. and J.S.; Writing–review & editing, J.Y. All authors have read and agreed to the published version of the manuscript.

**Funding:** This work was supported by the National Natural Science Foundation of China (grant number 51901181); the Young Talent fund of University Association for Science and Technology in Shaanxi, China (grant number 20200427); and the Postgraduate Innovation and Practice Ability Development Fund of Xi'an Shiyou University (grant number YCS22213133).

**Institutional Review Board Statement:** Not applicable.

**Informed Consent Statement:** Not applicable.

**Data Availability Statement:** The data presented in this study are available on request from the corresponding author after obtaining permission from the authorized individual.

**Conflicts of Interest:** The authors declare no conflict of interest.

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
