# Peer review of "Multi-Scale Structural Design and Advanced Materials for Thermal Barrier Coatings with High Thermal Insulation: A Review"

_coatings, doi:10.3390/coatings13020343_

Round 1

Reviewer 1 Report

In the present study, a detailed review on the thermal barrier coatings was given. The study is well organized and the data is presented properly. As it is well-known, review studies present future insights as well as conclusions. Therefore, it is recommended that a future insight paragraph should be added to the end of the conclusions.

Author Response

Dear reviewer,

Thank you very much for your valuable comments. We have carefully revised them in the revised manuscript according to your comments and marked them in red.

Point 1: In the present study, a detailed review on the thermal barrier coatings was given. The study is well organized and the data is presented properly. As it is well-known, review studies present future insights as well as conclusions. Therefore, it is recommended that a future insight paragraph should be added to the end of the conclusions.

Response 1: Thank you for your thoughtful suggestion. We added the outlook at the end of the revised manuscript, as follows:

…Thermal barrier coating technology is an indispensable technology for manufacturing high performance engines in the future. In recent years, researchers have made significant progress in the research of thermal barrier coating preparation technology, but there are still problems in the contradiction between coating performance and equipment cost and process cost, as well as the limitation of application scope. Future research should focus on the following areas: (i) Improved PS-TBC deposition process and post-treatment process to achieve porous TBC structures with segmented cracks. Enhanced thermal insulation and strain tolerance of the coating to increase the service life of TBCs. (ii) Diversified composite TBC preparation processes, such as Plasma Enhanced Chemical Vapor Deposition (PE-PVD) and laser CVD, to improve production efficiency and reduce process costs. (iii) Exploring new rare earth doped ceramic material systems and nanoscale ceramic material sizes to improve thermal insulation and bonding of coatings…

Reviewer 2 Report

The Authors of the manuscript „Multi-scale structural design and advanced materials for thermal barrier coatings with high thermal insulation: a review” undertake the interesting subject. The work does not discuss all the main issues related to TBC. But the topic of TBC is very broad and it is impossible to indicate all the issues. The manuscript is clearly presented and well organized. The investigation methodology is consistent with this type of investigation, and there are some useful results which can form the basis for publication. I feel that the paper deserves publication.

But the paper needs some corrections:

1)        There are some syntax and typing errors in the text. Please revise it carefully.

Author Response

Dear reviewer,

Thank you very much for your valuable comments. We have revised the manuscript by taking account of your suggestions. All the corrections have been highlighted by red colour. The following are responses and explanations to your comments.

Comments-in-brief: The Authors of the manuscript “Multi-scale structural design and advanced materials for thermal barrier coatings with high thermal insulation: a review” undertake the interesting subject. The work does not discuss all the main issues related to TBC. But the topic of TBC is very broad and it is impossible to indicate all the issues. The manuscript is clearly presented and well organized. The investigation methodology is consistent with this type of investigation, and there are some useful results which can form the basis for publication. I feel that the paper deserves publication.

Response to the comments-in-brief: Thank you for your positive evaluation on our work.

Point 1: There are some syntax and typing errors in the text. Please revise it carefully.

Response 1: Thank you for your thoughtful suggestion. We have refined the writing in the revised manuscript.  

Author Response

Dear reviewer,

Thank you very much for your valuable comments. We have revised the manuscript by taking account of your suggestions. All the corrections have been highlighted by red colour. The following are responses and explanations to your comments.

Comments-in-brief: The paper contains a review of the literature on the technology and materials used for TBCs. In the work the influence of various factors such the coating thickness, microstructure on their thermal insulating properties was also analysed. The paper is interesting. However, it is a pity that the authors only list the commonly used APS and PVD methods for obtaining the top coat for TBCs.

Response to the comments-in-brief: Thank you for your positive evaluation on our work.

Point 1: What do authors think about possibility of using other methods for this purpose (for example low temperature modifications of CVD such as MOCVD)?

Response 1: Thank you for your valuable suggestion. Compared with APS and EB-PVD, CVD can achieve uniform deposition of coatings on complex workpiece shapes and is suitable for depositing aluminide coatings on the inner cavity of turbine blades, resulting in smooth, dense and strong bonding surfaces. This process is the main method for the preparation of low activity aluminized coatings and platinum aluminum coatings. The biggest shortcoming of conventional CVD technology for TBC preparation is the low deposition rate of about 0.5-20 μm/min. PE-CVD and laser CVD were developed to overcome this shortcoming [1,2]. We added the outlook of CVD preparation TBC at the end of the revised manuscript.

[1] Goto, T. Thermal barrier coatings deposited by laser CVD. Coat. Technol. 2005, 198, 367-371, doi:https://doi.org/10.1016/j.surfcoat.2004.10.084.

[2] Préauchat, B.; Drawin, S. Properties of PECVD-deposited thermal barrier coatings. Coat. Technol. 2001, 142-144, 835-842, doi:https://doi.org/10.1016/S0257-8972(01)01211-7.

Point 2: The caption under figure (Figure 11) should be on the same page as the figure.

Response 2: Thank you for your thoughtful suggestion. We refined the writing in the revised manuscript.

Reviewer 4 Report

This paper gives an extensive review on the thermal barrier coatings.

The paper can be accepted if the authors incorporate the following queries.

1. The porosity of the coatings decreases with increase in coating thickness. this has to be addressed and explained.

2. It is stated that FGC improves the fracture toughness near the TC/BC interface, which prevents crack extension caused by the greater tip strain energy than the fracture toughness of the material under a thermomechanical load. how the fracture toughness is measured for the coatings.

Author Response

Dear reviewer,

Thank you very much for your valuable comments. We have revised the manuscript by taking account of your suggestions. All the corrections have been highlighted by red colour. The following are responses and explanations to your comments.

Reviewer 5 Report

In the manuscript, Song et al. review various aspects of the thermal insulation performance of thermal barrier coatings. The review is very well written, pleasant to read, and addresses the state-of-the-art regarding recent advances in improving the thermal insulation and lifetime of such coatings. There are some minor points that escaped the authors’ attention, and that can be clarified in the minor revision I am suggesting.

·         Line 34-37. Please rewrite (split) this sentence to make it easier to read.

·         Line 82-83. The font in figure 3 is too small for reading. Please, consider modifying it.

·         Line 129-130. Do “TC 129 280 +BC 100 μm” mean “…280 μm +..”? The same applies to “TTBC TC 1000 +BC 100 μm”.

·         Line 156. Please use the K unit instead of °C to be consistent with the previously discussed results for the reader's convenience.

·         Line 172. Commas are missing in the caption for Figure 9.

·         Line 394. Please change "resistant" to "resistance".

·         Line 504-508. In the caption for Figure 26 authors use “2.7 cracks mm-1” (and so on), which is unclear. Using “2.7 per mm” or the word density is recommended.

·         Line 517. “…plume with a diameter of...”

·         Line 633-634. The sentence seems to be unfinished.

·         Line 655. Please replace the word "bound"  with "bond" (...stronger bond strength").

·         Line 678. Figure 38: it is recommended, for better readability, to reduce the size of the (a) image and increase the size of the (b) one.

·         Line 705 and 727. This abbreviation “Sub” was not mentioned above in the text.

·         Line 762-763. Please, rewrite the sentence since it looks incomplete, particularly the part "...led to their from lamellar zones..."

·         Line 779-800. The caption for Figure 45 does not align perfectly with the corresponding Figure. Please modify it.

·         Line 823. "Resulting” is better to be changed to "results".

Author Response

Dear reviewer,

Thank you very much for your valuable comments. We have revised the manuscript by taking account of your suggestions. All the corrections have been highlighted by red colour. The following are responses and explanations to your comments.

Comments-in-brief: In the manuscript, Song et al. review various aspects of the thermal insulation performance of thermal barrier coatings. The review is very well written, pleasant to read, and addresses the state-of-the-art regarding recent advances in improving the thermal insulation and lifetime of such coatings. There are some minor points that escaped the authors’ attention, and that can be clarified in the minor revision I am suggesting.

Response to the comments-in-brief: Thank you for your positive evaluation on our work.

Point 1: Line 34-37. Please rewrite (split) this sentence to make it easier to read.

Response 1: Thank you for your thoughtful suggestion. We have refined the writing in the revised manuscript, as follows:

…Typical TBCs consists of a topcoat (TC), a bond coat (BC), a thermally grown oxide (TGO) and a substrate (SUB). The function of the TC are thermal insulation and resistance to calcium–magnesium–alumina–silicate (CMAS) corrosion. The function of the BC are bond strengthening, oxidation resistance and mechanical property transition…

Point 2: Line 82-83. The font in figure 3 is too small for reading. Please, consider modifying it.

Response 2: Thank you for your valuable suggestion. We have increased the font size in the revised manuscript, as follows:

Point 3: Line 129-130. Do “TC 129 280 +BC 100 μm” mean “…280 μm +..”? The same applies to “TTBC TC 1000 +BC 100 μm”.

Response 3: Thank you for your valuable suggestion. That's what it means. We have refined the writing in the revised manuscript, as follows:

…TBC (TC 280 μm +BC 100 μm) and a thick TBC (TTBC TC 1000 μm +BC 100 μm)…

Point 4: Line 156. Please use the K unit instead of °C to be consistent with the previously discussed results for the reader's convenience.

Response 4: Thank you for your valuable suggestion. We have refined the writing in the revised manuscript, as follows:

…250 μm thick TBCs can reduce the surface temperature of the blade substrate by 384 K–440 K…

Point 5: Line 172. Commas are missing in the caption for Figure 9.

Response 5: Thank you for your valuable suggestion. We have refined the writing in the revised manuscript, as follows:

…(Htc is thickness of top coat, LE, LT, TT are leading edge, leading tip, and trailing tip, respectively).

Point 6: Line 394. Please change "resistant" to "resistance".

Response 6: Thank you for your valuable suggestion. We have refined the writing in the revised manuscript, as follows:

…indicating favorable delamination resistance and long lifetime…

Point 7: Line 504-508. In the caption for Figure 26 authors use “2.7 cracks mm-1” (and so on), which is unclear. Using “2.7 per mm” or the word density is recommended.

Response 7: Thank you for your valuable suggestion. We have refined the writing in the revised manuscript, as follows:

…(a), (b) and (c) 2.7 mm−1 crack density coating cycled to 1238 °C for 1200 cycles and 1770 cycles, and to 1335 °C for 320 cycles, respectively; (d), (e) and (f) 2.7 mm−1 crack density coating cycled to 1226 °C for 1650 cycles and 1810 cycles, and to 1317 °C for 174 cycles, respectively; (g) and (h) 0.9 mm−1 crack density coating cycled to 1216 °C for 1071 cycles and to 1327 °C for 217 cycles, respectively…

Point 8: Line 517. “…plume with a diameter of...”

Response 8: Thank you for your valuable suggestion. PS-PVD plasma plume diameter of 200−400 mm.

Point 9: Line 633-634. The sentence seems to be unfinished.

Response 9: Thank you for your valuable suggestion. We have refined the writing in the revised manuscript, as follows:

…The selection of appropriate powder particle size and spraying parameters can improve the bond strength…

Point 10: Line 655. Please replace the word "bound"  with "bond" (...stronger bond strength").

Response 10: Thank you for your valuable suggestion. We have replaced it in the revised manuscript.

Point 11: Line 678. Figure 38: it is recommended, for better readability, to reduce the size of the (a) image and increase the size of the (b) one.

Response 11: Thank you for your valuable suggestion. We have modified it in the revised manuscript.

Point 12: Line 705 and 727. This abbreviation “Sub” was not mentioned above in the text.

Response 12: Thank you for your valuable suggestion. We have refined the writing in the revised manuscript, as follows:

…The bottom layer of YSZ enhances the fracture toughness of the coating and provides a transition adjustment for the thermal expansion mismatch between the advanced material and substrate…

…which overcame the thermal mismatch between the LMA TC and substrate, have better strain tolerance and thermal cycling life than the single-layer YSZ and LMA coatings…

Point 13: Line 762-763. Please, rewrite the sentence since it looks incomplete, particularly the part "...led to their from lamellar zones..."

Response 13: Thank you for your valuable suggestion. We have refined the writing in the revised manuscript, as follows:

…Therefore, rapid densification of the nanozones led to the formation of coarse voids…

Point 14: Line 779-800. The caption for Figure 45 does not align perfectly with the corresponding Figure. Please modify it.

Response 14: Thank you for your valuable suggestion. We have modified it in the revised manuscript, as follows:

…Changes of (a) elastic modulus; (b) thermal conductivity as a function of thermal exposure duration…

Point 15: Line 823. "Resulting” is better to be changed to "results".

Response 15: Thank you for your valuable suggestion. We have modified it in the revised manuscript, as follows:

…the TBC thickness increased by 0.2 mm, results in a decrease in the maximum blade temperature of approximately 40 K…
